# In-situ cryo-immune engineering of tumor microenvironment with cold-responsive nanotechnology for cancer immunotherapy

Wenquan Ou[1], Samantha Stewart[1], Alisa White[1], Elyahb A. Kwizera[1], Jiangsheng Xu[1], Yuanzhang Fang[2], James G. Shamul[1], Changqing Xie [3], Suliat Nurudeen[4], Nikki P. Tirada[4], Xiongbin Lu [2], Katherine H. R. Tkaczuk [4] & Xiaoming He [1,4] ✉

Cancer immunotherapy that deploys the host's immune system to recognize and attack tumors, is a promising strategy for cancer treatment. However, its efficacy is greatly restricted by the immunosuppressive (i.e., immunologically cold) tumor microenvironment (TME). Here, we report an in-situ cryo-immune engineering (ICIE) strategy for turning the TME from immunologically "cold" into "hot". In particular, after the ICIE treatment, the ratio of the CD8+ cytotoxic T cells to the immunosuppressive regulatory T cells is increased by more than 100 times in not only the primary tumors with cryosurgery but also distant tumors without freezing. This is achieved by combining cryosurgery that causes "frostbite" of tumor with cold-responsive nanoparticles that not only target tumor but also rapidly release both anticancer drug and PD-L1 silencing siRNA specifically into the cytosol upon cryosurgery. This ICIE treatment leads to potent immunogenic cell death, which promotes maturation of dendritic cells and activation of CD8+ cytotoxic T cells as well as memory T cells to kill not only primary but also distant/metastatic breast tumors in female mice (i.e., the abscopal effect). Collectively, ICIE may enable an efficient and durable way to leverage the immune system for combating cancer and its metastasis.

Cancer is the second leading cause of death globally and metastatic cancer is the major reason for cancer-related mortality[1,2]. Harnessing the immune system to battle cancer via immunotherapy has emerged as a powerful and potentially revolutionizing strategy for treating cancer metastasis[3,4]. Although therapeutic advances with tumor antigen vaccines, chimeric antigen receptor (CAR) T cells, and other cancer immunotherapy strategies have shown promising success in both pre-clinical and clinical studies of hematological malignancies, their efficacy of destroying solid tumors in most cancer patients has been limited[4–7]. The immunosuppressive (i.e., immunologically cold) tumor microenvironment (TME) including malignant cells, deactivated/compromised immune cells, and soluble factors, is a key factor that contributes to the poor clinical outcomes of immunotherapy of solid tumors[8–13].

Cryosurgery has been proposed as a promising strategy to modulate the TME in cancer treatment[14–16]. Cryosurgery is done by cooling to cause ice formation (i.e., frostbite) in tumor, which leads to cryoinjury and cancer cell death inside the frozen tissue iceball[17–20]. This may result in the release of tumor antigens and the production of damage-associated molecular patterns (DAMPs)[14,21–23]. These

[1]Fischell Department of Bioengineering, University of Maryland, College Park, MD 20742, USA. [2]Department of Medical and Molecular Genetics and Melvin and Bren Simon Cancer Center, Indiana University School of Medicine, Indianapolis, IN 46202, USA. [3]Thoracic and Gastrointestinal Malignancies Branch, Center for Cancer Research, National Cancer Institute, National Institutes of Health, Bethesda, MD 20892, USA. [4]Marlene and Stewart Greenebaum Comprehensive Cancer Center, University of Maryland, Baltimore, MD 21201, USA. ✉e-mail: shawnhe@umd.edu

molecules may provoke immunogenic cell death (ICD), shape an immuno-active (i.e., immunologically hot) TME, and further stimulate dendritic cells (DCs), macrophages, as well as CD8[+] cytotoxic T lymphocytes (CTLs) to execute antitumor immunotherapy[21,23–26]. In addition, unlike high-temperature thermal therapy for which vascular stasis occurs during heating, there is a temporary (a few hours) reperfusion of the tumor immediately after thawing a frozen tumor iceball[27–29], which may allow efficient immune cell infiltration into the tumor after cryosurgery. However, cryosurgery alone is insufficient to stimulate a potent immunotherapeutic effect against cancer.

Cryosurgery has been used clinically and become increasingly popular for treating cancers, particularly for breast cancer. This is because it is minimally invasive with minimized cosmetic damage compared to conventional surgery with a scalpel, and the frozen tumor iceball can be conveniently monitored in real-time using medical ultrasonography due to the hyperechoic nature of ice[17,30–35]. Unfortunately, a temperature of −20 °C or below is required to ensure cell death while the temperature in the peripheral region of a frozen tumor iceball detectable by medical ultrasonography is above −20 °C and up to ~−4 °C[17–20,36,37]. The latter is because biological tissues freeze gradually with the decrease of temperature starting from ~−0.6 °C to below −20 °C during cryosurgery[20,37,38], and the extent of ice formation is insufficient for detection (based on the aforementioned hyperechoic property of ice) by medical ultrasonography until ~−4 °C. In other words, medical ultrasonography can neither detect ice formation above ~−4 °C nor tell where the temperature is below −20 °C that is needed to ensure cancer cell death. This may lead to incomplete tumor

destruction and cancer recurrence after cryosurgery[17,30–33]. Therefore, combining cryosurgery with other cancer therapies has been explored to improve cancer-killing in the peripheral region of a frozen tumor iceball for enhanced therapeutic outcome[39–46]. However, despite the efficacy of destroying localized tumors with the combination therapies, their capability of effectively killing distant and metastatic tumors (i.e., the abscopal effect of cryoimmunotherapy) has not been reported in the literature. Furthermore, no work has been done to utilize nanotechnology (particularly the one that is responsive to the cold/freezing temperature during cryosurgery) for delivering immunotherapy to combine with cryosurgery for enhancing the efficacy and safety of cancer therapy.

Here we report an in-situ cryo-immune engineering (ICIE) approach to turn the TME from immunologically cold into hot, by developing a type of cold-responsive nanoparticles (CRNPs) to co-deliver chemotherapy and immunotherapy agents for combining with cryosurgery (Fig. 1). This not only greatly enhances cancer cell killing at the edge of a frozen tumor iceball with a high subzero temperature (from ~−4 to −20 °C) that alone can not kill cancer cells effectively, but also stimulates cryo-immunotherapy against both the primary tumor with the cryosurgery treatment and distant/metastatic tumors with no cryosurgery. Taken together, ICIE may be a strategy for reversing the immunosuppressive TME to generate potent tumor-specific immune responses against not only primary but also distant and metastatic tumors with no evident side effects. This is of great significance as cancer metastasis is the major cause of most cancer-related mortality.

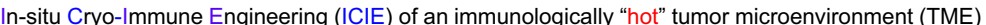

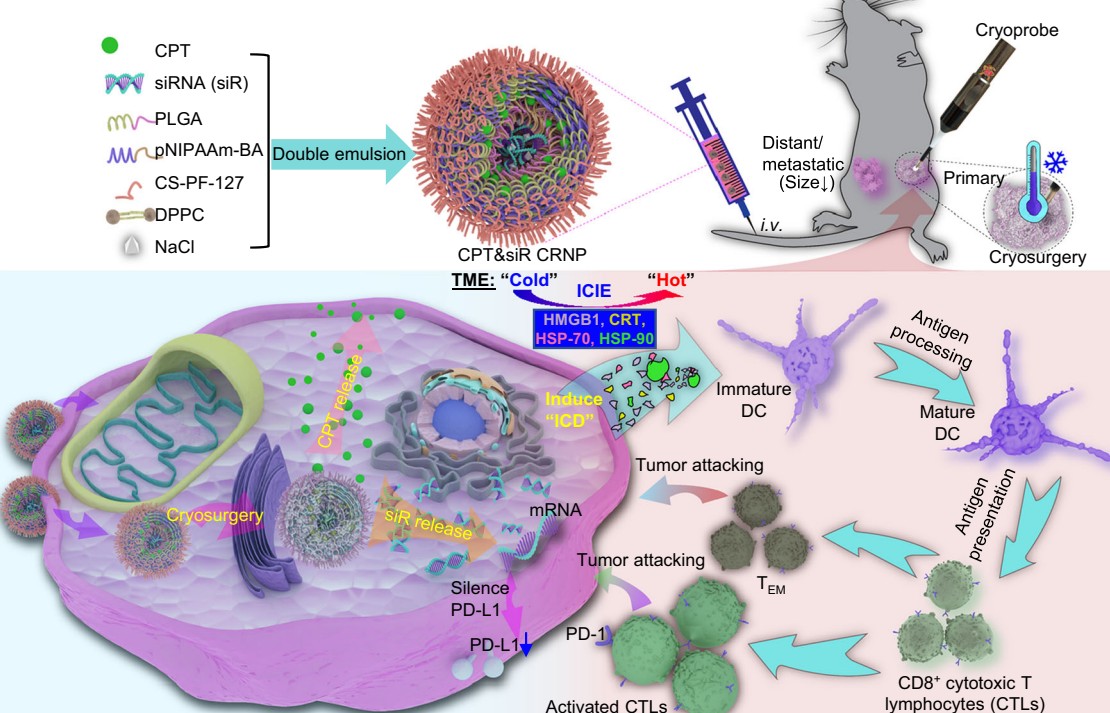

**Fig. 1 | A schematic illustration of the ICIE strategy for killing both primary and distant/metastatic tumors.** ICIE combines cryosurgery with intravenously injected (i.v.) irinotecan (or camptothecin/CPT) and PD-L1 silencing siRNA (siR)-laden cold-responsive nanoparticles (CPT&siR CRNPs) to engineer the tumor microenvironment (TME), via promoting immunogenic cell death (ICD) and reducing the expression of programmed death Ligand 1 (PD-L1) in cancer cells. This turns the TME from immunologically "cold" (i.e., immunosuppressive) into immunologically "hot" (i.e., immuno-active). As a result, the CD8[+] T cells can be activated to exert

tumor eradication both locally and systemically. The CPT&siR CRNPs are prepared by a double-emulsion method detailed in the Methods section using CPT, siR, Poly (D, L-lactide-co-glycolide) (PLGA), poly (N-isopropylacrylamide copolymerized butyl acrylate) (pNIPAAm-BA), chitosan-modified PF-127 (CS-PF-127), 1,2-dipalmitoyl-sn-glycero-3-phosphocholine (DPPC), and sodium chloride (NaCl). HMGB1: high mobility group box protein 1, CRT: calreticulin, HSP-70: heat shock protein-70, HSP-90: heat shock protein-90, DC: dendritic cell, and T_EM: effective memory T cell.

## Results

### Synthesis and characterization of CRNPs for cold-triggered drug and gene release

Previous studies used polymers with a lower critical solution temperature (LCST) of -14−30 °C to synthesize thermally responsive nanoparticles[47,48], which can not be used to precisely control the drug and gene release within the frozen tumor iceball while sparing the surrounding normal tissue in response to the cold temperature (-−4 °C) at the outer surface of an iceball during cryosurgery. Therefore, we synthesized a series of poly (N-isopropylacrylamide copolymerized with butyl acrylate) (pNIPAAm-BA) polymers with varying LCSTs by adjusting the feeding ratio of NIPAAm to BA. Proton nuclear magnetic resonance (¹H-NMR) spectroscopy analyses of chemical bonds show successful synthesis of the pNIPAAm-BA polymers (Supplementary Fig. 1a). The number-averaged molecular weight ($M_n$) ranges from -67 to 83 kDa as determined by gel permeation chromatography (GPC, Supplementary Fig. 1b, c). The LCST of the polymers dispersed in different solutions was measured with cryo-microscopy. As shown in Supplementary Fig. 1d, the LCST of the same polymer displays no significant variation in different solvents/solutions, but a higher NIPAAm content leads to a higher LCST of the pNIPAAm-BA polymer. The resultant p(NIPAAm)$_{589}$-co-(BA)$_{117}$, with an LCST of −4.4 ± 0.6 °C that is close to the temperature on the outer surface of a frozen tumor iceball, was selected to synthesize the CRNPs for co-encapsulating irinotecan (CPT, a clinically used chemotherapy drug) and programmed death-ligand 1 (PD-L1) silencing siRNA (siR) using a double-emulsion method (Fig. 1). Other materials used for synthesizing the CRNPs include poly (D, L-lactide-co-glycolide) (PLGA), Pluronic F127 (PF127), 1,2-dipalmitoyl-sn-glycero-3-phosphocholine (DPPC), and chitosan. Most of the materials are either FDA-approved for medical use (PLGA, PF127, and DPPC) or naturally derived material (chitosan) with good biocompatibility[49–51], which may facilitate the clinical translation of the CRNPs. The surface of the resultant CPT&siR-laden CRNPs (or CPT&siR CRNPs for short) is decorated with chitosan (CS) for enhanced tumor-targeting capability[50].

To optimize the loading of CPT and siR in the CRNPs, the encapsulation efficiency (EE) and loading content (LC) of CPT were measured for feeding CPT at 1, 3, and 5% of the polymers. As shown in Supplementary Fig. 2a, b, the EE of CPT decreases monotonically with the increase of the CPT feeding percentage, while the LC of CPT reaches a plateau at 3% feeding percentage. Therefore, the 3% CPT feeding percentage was used in this work, for which the LC of CPT is 1.5 ± 0.2% with an encapsulation efficiency (EE) of 51.2 ± 8.2%. For siR, the EE and LC are 73.4 ± 4.2% and 0.14 ± 0.01%, respectively. The particle size of the optimized CPT&siR CRNPs is 143.6 ± 6.6 nm in diameter with a surface zeta potential of −4.9 ± 0.4 mV. Transmission electron microscopy (TEM) analysis of the CPT&siR CRNPs shows that they have a core-shell structure (at 22 °C with no cooling, Fig. 2a). Furthermore, they are stable over time and show negligible changes in their size after being suspended in phosphate-buffered saline (PBS) for five days (Supplementary Fig. 2c). Moreover, compared to the free siR that easily gets degraded within 5 minutes of incubation with PBS at 22 °C (Supplementary Fig. 2d), the siR encapsulated in the CPT&siR CRNPs remains stable for at least 24 h (Supplementary Fig. 2e).

The cold-responsiveness of the CPT&siR CRNPs was studied by TEM and scanning electron microscopy (SEM) first. As shown in the TEM and SEM images in Fig. 2a, destruction of the structure of the CRNPs after cold treatment (incubating the samples at −4 °C for 10 min and warming back to 22 °C) is evident. The cold-induced destruction of the CRNP structure can trigger the release of the encapsulated siR, resulting in an evident band (white) of siR in agarose gel for the CPT&siR CRNPs with the cold treatment at 4 °C (Supplementary Fig. 3a). The hydrodynamic diameter of the CPT&siR CRNPs is greatly increased with two wide peaks (one at -700 nm and the other at -5000 nm on average) after the cold treatment (Fig. 2b), probably due

to aggregation of the hydrophobic polymers/drugs dissembled from the CPT&siR CRNPs. According to the photographs and cryo-microscopy images of the nanoparticle samples in deionized water (DW, 12.0 mg ml⁻¹, Fig. 2c), the aqueous sample of CPT&siR-laden poly (D, L-lactic-co-glycolic acid) (PLGA) nanoparticles (CPT&siR PLGA NPs, 135.7 ± 6.1 nm in diameter with a surface zeta potential of −14.1 ± 0.6 mV) that are not cold-responsive, has a homogeneous and milky appearance and blocks light to give a dark cryo-microscopic image at both 22 °C (either before or after cooling) and −4 °C. In contrast, a homogeneous and transparent aqueous sample can be observed for the CPT&siR CRNPs at −4 °C, although the sample is also homogeneous and milky at 22 °C before the cooling treatment. The cryo-microscopic images showing the cold-triggered changes were further analyzed to quantify their grayscale intensity. When slowly cooled (1 °C min⁻¹) below -4 °C, the sample of the CPT&siR CRNPs becomes more and more transparent with lower and lower grayscale intensity (Fig. 2d). In comparison, the grayscale intensity of the microscopic images for the sample of the CPT&siR PLGA NPs remained constant during cooling. When warmed back to 22 °C, the sample with CPT&siR CRNPs becomes heterogeneous with a large whitish aggregate of insoluble materials piling up at the bottom (Fig. 2c). These results confirm the cold-triggered dissolution and irreversible disassembly of the CPT&siR CRNPs.

To determine the cold-triggered drug release profile, the CPT&siR CRNPs were incubated in different buffers at pH 5.0, 6.5, and 7.4 to mimic that in lysosomes, the extracellular space of tumor, and normal tissue, respectively. Before applying the cold treatment, both CPT and siR are released in a sustained manner at 37 °C, with less than 10% release in 8 h (Supplementary Fig. 3b–d). After cold treatment by incubating the samples at −4 °C for 10 min, cold-triggered rapid release profiles are observable for both CPT and siR in all three different buffers, due to the cold-induced disassembly of the CRNPs (Fig. 2a, c).

### Cold-triggered endo/lysosomal escape of siR and enhancement of PD-L1 silencing

To investigate cellular uptake and intracellular trafficking of the CRNPs, CRNPs loaded with CPT and cyanine 5 (Cy5)-labeled siR were incubated with green fluorescent protein-positive EO771 (GFP⁺ EO771) breast cancer cells and lymphocytes from mouse spleen at 37 °C for 8 h. Significant red fluorescence signal of Cy5 can be observed in the GFP⁺ EO771 cells with confocal microscopy, indicating successful uptake of the CPT&Cy5-siR CRNPs by the cells (Supplementary Fig. 4a). In contrast, negligible Cy5 fluorescence could be detected in the cells incubated with free CPT and Cy5-siR, probably due to degradation and/or poor uptake of the siR. Successful cellular uptake of the CPT&Cy5-siR CRNPs is confirmed by flow cytometry, which shows over 90% of the GFP⁺ EO771 cells are Cy5 positive (Supplementary Fig. 4b, c). For non-cancerous cells like lymphocytes isolated from the spleen, only -0.9% of them are Cy5 positive (Supplementary Fig. 4d), suggesting the good cancer cell targeting capability of the CRNPs. We further investigated if cold treatment of CRNPs in cells can trigger endo/lysosomal escape of the siR into the cytosol. Without cold treatment (CPT&Cy5-siR CRNPs), Cy5 fluorescence can be observed in the endo/lysosomes of EO771 cells after 8 h of incubation (Fig. 2e), indicated by the evident overlap of Cy5 (red) and endo/lysosomes (green) as shown by the confocal images, the distribution of the red and green fluorescence intensities, and the high intensity of Cy5-siR fluorescence in the endo/lysosomal areas. After cold treatment (CPT&Cy5-siR CRNPs+C), colocation of Cy5 and endo/lysosomes is decreased with an evident separation of red and green fluorescence signals, indicating successful cold-triggered escape of Cy5-siR from endo/lysosomes into the cytosol. The cold-triggered enhancement of endo/lysosomal escape is due to the encapsulation of sodium chloride (NaCl) in the CRNPs. In EO771 cells treated with the CRNPs containing

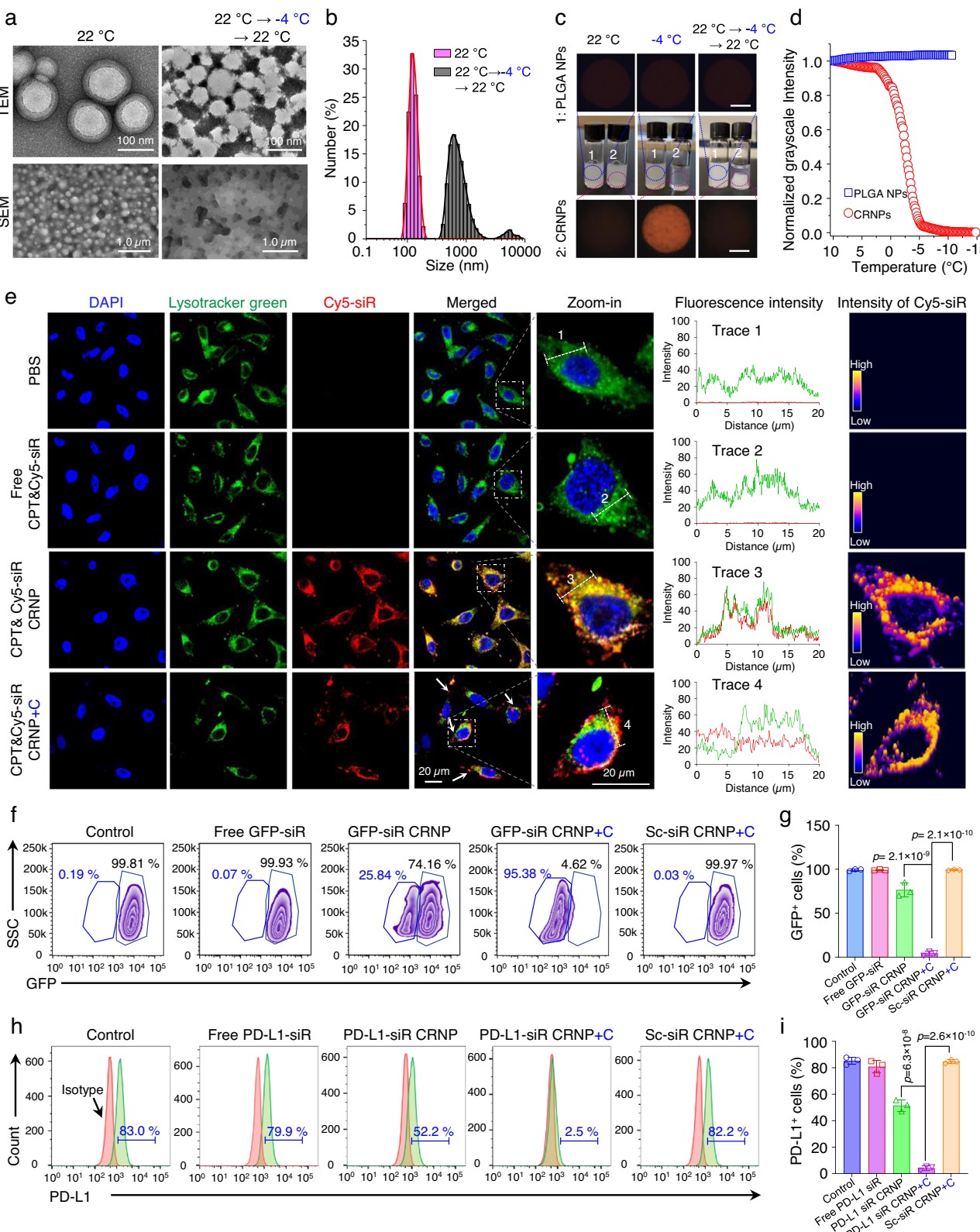

no NaCl, cold treatment (CPT&Cy5-siR CRNPs+C, no NaCl) does not induce an evident separation between the red fluorescence of Cy5-siR and the green fluorescence of endo/lysosomes (Supplementary Fig. 5). The underlying mechanism for NaCl-induced endo/lysosomal escape may be ascribed to the rapid release of the encapsulated $Na^+$ and $Cl^-$ into the endo/lysosomes triggered by cold treatment, resulting in a surge of osmolality (hypertonicity) and influx of water to destabilize or rupture endo/lysosomes[52]. As a result, the encapsulated CPT and PD-L1

silencing siRNA are rapidly released into the cytosol to perform the chemotherapy and gene silencing functions.

To investigate the gene silencing effect of CRNPs-delivered siR, we incubated GFP⁺ EO771 cells with CRNPs containing GFP silencing siRNA (GFP-siR CRNPs) for 8 h, cold-treated the cells at −4 °C for 10 min, and further cultured them for 40 h. The cold-triggered GFP-siR release from the CRNPs markedly decreases the expression of GFP in the GFP⁺ EO771 cells, according to the data of confocal microscopy

**Fig. 2 | Cold-responsiveness of CRNPs for enhanced endo/lysosomal escape of siRNA and gene silencing. a** Transmission electron microscopy (TEM) and scanning electron microscopy (SEM) images of CPT&siR CRNPs before and after cold treatment at −4 °C for 10 min. **b** Hydrodynamic size (diameter) distribution of the CPT&siR CRNPs before and after the cold treatment. **c** Typical photographs and cryo-microscopy images of CPT&siR PLGA NPs that are not cold-responsive and CPT&siR CRNPs at different temperatures, showing the cooling-enhanced transparency of the aqueous suspension of the CRNPs due to their disassembly at cold temperature. Scale bar: 500 μm. **d** Average grayscale intensity of the aqueous samples of CPT&siR PLGA NPs and CPT&siR CRNPs from their cryo-microscopy images taken at different temperatures to show the cold-responsiveness of the CPT&siR CRNPs. **e** Confocal images (left 5 columns), fluorescence intensity distribution (column 6), and intensity distribution of Cy5-siR (right column) of EO771 cells after incubation with PBS, free CPT&Cy5-siR, CPT&Cy5-siR CRNPs, and CPT&Cy5-siR CRNPs+C for 8 h. "+C" indicates cold treatment at −4 °C for 10 min

done after the 8 h incubation. Blue: 4′,6-diamidino-2-phenylindole (DAPI bound to DNA in cell nuclei, excited at 405 nm that does not excite CPT fluorescence), red: cyanine 5 labeled siRNA (Cy5-siR), green: LysoTracker Green. **f, g** Expression of GFP fluorescence characterized by flow cytometry showing typical flow cytometry histograms (**f**) and the quantitative data (**g**) of GFP+ EO771 cells either with no treatment or after treated with free GFP-siR, GFP-siR CRNPs, GFP-siR CRNPs+C, and Sc-siR CRNPs+C (n = 3 independent experiments). GFP: green fluorescence protein. Sc-siR: scrambled siRNA, +C: with cold treatment at −4 °C for 10 min. **h, i** Expression of PD-L1 analyzed by flow cytometry showing typical flow cytometry histograms (**h**) and the quantitative data (**i**) of EO771 cells after different treatments (n = 3 independent experiments). Statistical analyses were done using one-way analysis of variance (ANOVA) with Tukey's multiple comparisons test and correction. The experiments for **a**–**e** were repeated three times independently with similar results. Data are presented as mean ± SD (**g**, **i**). Source data are provided as a Source Data file.

(Supplementary Fig. 6) and flow cytometry (Fig. 2f–g). Without cold treatment, GFP fluorescence is eliminated in ~26% of GFP+ EO771 cells treated with the GFP-siR CRNPs. After cold treatment of the cells incubated with GFP-siR (CRNPs+C), the percent of cells with silenced GFP expression increases to ~95%, indicating cold-triggered siR release and its capability of effectively silencing GFP expression in the cells. This is further confirmed by treating EO771 cells with CRNPs loaded with siR specific for silencing PD-L1. Cold-triggered release of the PD-L1 silencing siR from CRNPs (PD-L1-siR CRNPs+C) results in a much lower expression of PD-L1 in EO771 cells (2.5%) than the treatments of free PD-L1 siR and PD-L1-siR CRNPs without cooling (Fig. 2h, i). These results indicate that the cold-triggered siR release from CRNPs significantly enhances their gene-silencing ability.

To evaluate the anticancer capacity of CPT&siR CRNPs, EO771 cells were incubated with blank CRNPs, free CPT&siR, and CPT&siR CRNPs at various CPT concentrations for 24 h with and without cold treatment. CPT&siR CRNPs show significantly higher efficacy to kill the cancer cells than free CPT&siR (Supplementary Fig. 7), probably due to poor cellular uptake of free CPT&siR. Importantly, cold treatment further significantly enhances the efficacy of CPT&siR CRNPs to kill cancer cells, showing the cold-triggered rapid release of CPT is more effective in killing cancer cells than the slow and sustained release of CPT from the nanoparticles. It is worth noting that the viability of cells treated with the blank CRNPs (up to 5.6 mg ml⁻¹) is higher than ~90%. The cancer cell killing efficacy of cold-treated CPT&siR CRNPs at a CPT concentration of 10.0 μg ml⁻¹ is further confirmed by flow cytometry data, showing a high percentage of apoptosis/necrosis (77.9%) in EO771 cells with the CPT&siR CRNPs+C treatment (Supplementary Fig. 8).

**ICIE increases the expression of DAMPs and maturation of DCs**
We next studied the impact of ICIE (i.e., the CPT&siR CRNPs+C treatment) on ICD via examining the expression of DAMPs and maturation of DCs. For cold treatment, EO771 cells were incubated with the CPT&siR CRNPs or other control formulations including PBS, free CPT&siR, CPT-laden CRNPs (CPT CRNPs), and PD-L1 silencing siR-laden CRNPs (siR CRNPs) for 8 h, cold-treated at −4 °C or −20 °C for 10 min, and further cultured for 16 h. Flow cytometry was used to quantify the expression of DAMPs in the cells, including heat shock protein-90 (HSP-90), heat shock protein-70 (HSP-70), calreticulin (CRT), and high mobility group box 1 protein (HMGB1)[14,23,24], following the different treatments. Cold treatments (at both −4 and −20 °C) significantly enhance the production of all the DAMPs for all the formulations, compared to 37 °C (Fig. 3a). These data indicate that cold treatment alone can promote DAMP expression and induce ICD. Particularly, the formulation of CPT&siR CRNPs with cold treatments (i.e., ICIE) at −4 and −20 °C induces higher expression of HSP-70, HSP-90, HMGB1, and CRT than all the other formulations with the same cold treatment, indicating that cold-triggered release of both CPT and

siR from the CPT&siR CRNPs helps boost ICD beyond the cold treatment alone. When the frozen tumor cells undergo ICD, the released DAMPs promote recruitment, accumulation, as well as the maturation of DCs in the TME, which should enhance the engulfment of tumor antigens by DCs to facilitate antigen presentation to T cells. This eventually should result in the activation of CD8+ CTLs to kill tumor cells[53,54], as schematically illustrated in Fig. 3b.

The activation of an antitumor immune response requires the maturation of DCs and presentation of tumor antigens to T cells. To this end, the maturation of DCs with the capability of activating CD8+ T cells (that can recognize the OVA antigen) was evaluated by co-culture of DCs and cancer cells (EO771-OVA) with various treatments. As shown in Fig. 3c, d, ICIE (CPT&siR CRNPs+C) treatment at −20 °C elevates the maturation of bone marrow dendritic cells (BMDCs, CD11c+CD86+) to 49.4%. This level is significantly higher than that for the CPT&siR CRNPs (no cold treatment, by default), free CPT&siR (no cold treatment, by default), and PBS (no cold treatment, by default). Antigen presentation capabilities in these mature BMDCs were examined by detecting the level of ovalbumin-derived peptide with the amino acid sequence of SIINFEKL that binds to the H-2Kb of major histocompatibility complex I (MHC I) molecule (to result in Kb-SIINFEKL) in antigen-presenting cells (APCs). BMDCs co-cultured with the EO771-OVA cells treated with ICIE express the highest amount of SIINFEKL (58.9%, Fig. 3e, f). Indeed, this is significantly higher than that for BMDCs co-cultured with the EO771-OVA cells treated with CPT&siR CRNPs, free CPT&siR, and PBS (31.2, 17.0, and 3.1%, respectively, Fig. 3e, f). These data indicate the enhanced maturation of DCs with the capability of activating CD8+ T cells that can recognize the OVA antigen. Moreover, the co-culture of CD8+ T cells and BMDCs matured by the EO771-OVA cells that received the ICIE treatment secretes 2.3-, and 1.6-fold more pro-inflammatory cytokines (IFN-γ and TNF-α) than the co-culture of CD8+ T cells and BMDCs matured by the EO771-OVA cells that received CPT&siR CRNPs without cold treatment (Supplementary Fig. 9a, b), suggesting the potentiation of antitumor immune response via increased DC maturation when combining cryosurgery with CPT&siR CRNPs.

**ICIE enhances the proliferation, activation, and tumor-attacking efficacy of CD8+ T cells**
When co-cultured with CD8+ T cells, BMDCs matured by EO771-OVA cells treated with ICIE at −20 °C substantially increase the proliferation of CD8+ T cells (78.9%, Fig. 3g, h), which is significant compared to the treatments of CPT&siR CRNPs, free CPT&siR, and PBS (34.9, 16.4, and 8.3%, respectively). To investigate the activation of CD8+ T cells, we examined the expression of cytotoxic granzyme B (GZMB, a serine protease commonly found in the granules of CTLs) using flow cytometry. Indeed, ICIE increases the percentage of CD8+GZMB+ to 56.5%, which is significantly higher than that for CPT&siR CRNPs, free CPT&siR, and PBS (36.1, 26.4, and 17.0%, respectively, Fig. 3i, j).

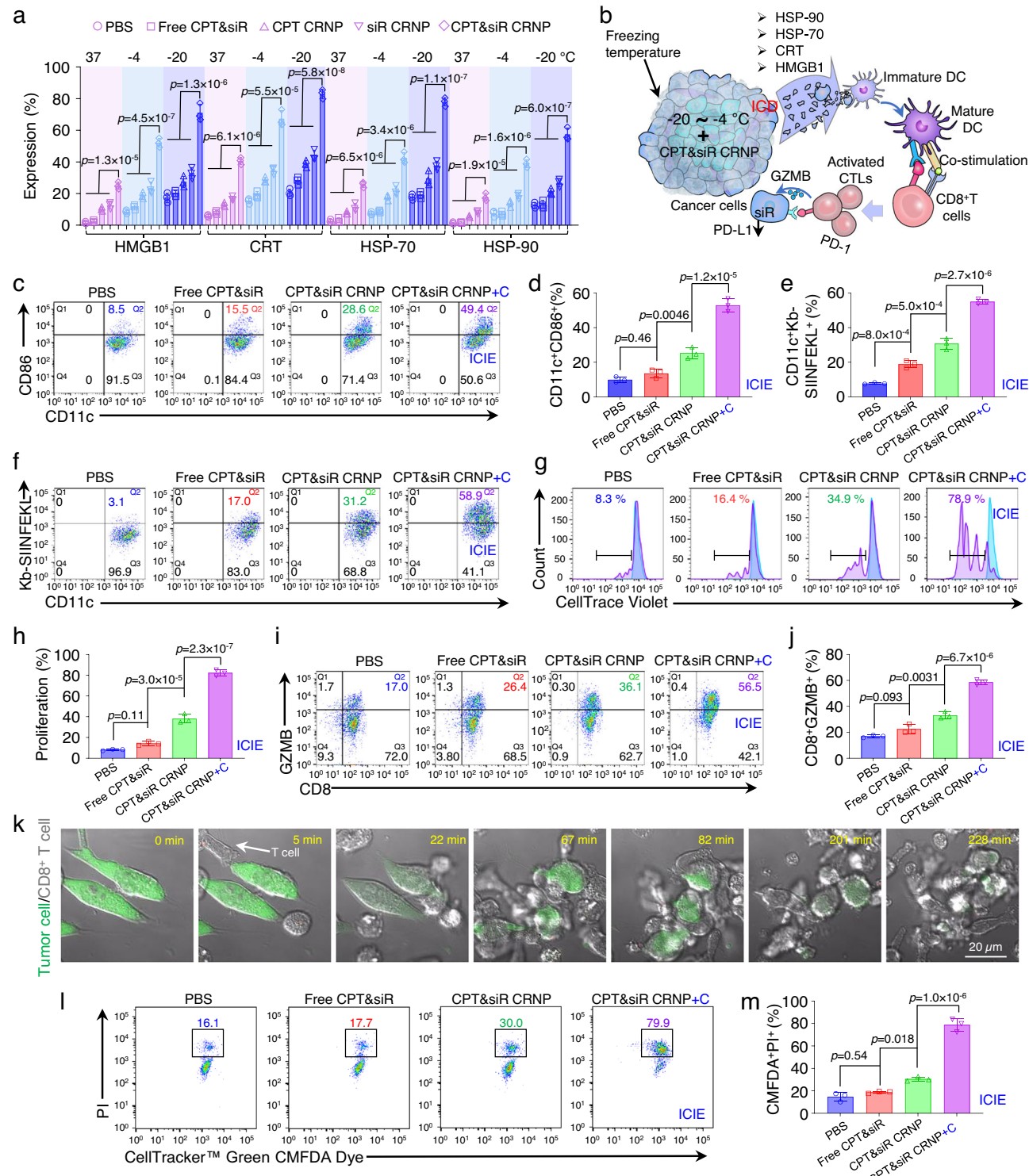

We next tested the tumor-attacking capability of the activated CD8+ T cells. CD8+ T cells activated by BMDCs that are matured by EO771-OVA cells receiving the ICIE treatment, migrate to and accumulate around the EO771-OVA tumor cells (labeled with green color) within 5 min to attack them persistently and the tumor cells are eventually dismantled within 4 h (Fig. 3k, Supplementary Movie 1). Furthermore, CD8+ T cells from the ICIE group result in a significantly higher percentage of cancer cell death (79.9%) than the cells from the CPT&siR CRNPs (30.0%) group (Fig. 3l, m). CD8+ T cells from the PBS and free CPT&siR groups cause 16.1% and 17.7% of EO771-OVA cell death, respectively. To assess the synergistic effects of co-delivery of

CPT and siR, we compared the cancer cell killing capability of CPT&siR CRNPs (either with or without cold treatment) with that CPT CRNPs alone, siR CRNPs alone, and the simple addition (i.e., additive effect) of the treatments with CPT CRNPs alone and siR CRNPs alone. In the absence of cold treatment, co-delivery of the two agents in the CPT&siR CRNPs shows no synergistic effect (Supplementary Fig. 10a, b): it leads to ~31% of cancer cell death, which is less than the percentage of cancer cell death (~39%) due to the additive effect of the CPT CRNPs alone and siR CRNPs alone. Importantly, when cold treatment is applied, the synergistic effect of CPT and siR in CPT&siR CRNPs is evident (Supplementary Fig. 10c, d): it causes a significantly higher

**Fig. 3 | Cryosurgery reinforces CPT&siR CRNPs in inducing T cell activation and tumor attacking via production of ICD. a** Expression of HMGB1, CRT, HSP-70, and HSP-90 in EO771 cells after incubating them with various formulations at −20, −4, or 37 °C for 10 min (*n* = 3 independent experiments). **b** A schematic illustration of ICD production following treatment of CPT&siR CRNPs in combination with cryosurgery to mature DCs, present the tumor-specific antigen to T cells by the matured DCs, and activate CD8[+] cytotoxic T lymphocytes (CTLs) to attack cancer cells with reduced expression of PD-L1 via the release of granzyme B (GZMB). **c**, **d** Typical flow cytometry plots (**c**) and quantitative data (**d**) on maturing bone marrow dendritic cells (BMDCs) after co-culturing them with EO771-OVA cells for 24 h (*n* = 3 independent experiments). The EO771-OVA cells were pretreated with the indicated various formulations. "+C" represents cold treatment at −20 °C for 10 min. **e**, **f** Quantitative data (**e**) and typical flow cytometry plots (**f**) of CD11c[+]Kb-SIINFEKL[+] BMDC percentage following co-culture with the aforementioned, pretreated EO771-OVA cells (*n* = 3 independent experiments). **g**, **h** Typical flow cytometry

histograms (**g**) and quantitative data (**h**) of CD8[+] T cell proliferation after incubating them with the BMDCs matured by the EO771-OVA cells treated with aforementioned formulations (*n* = 3 independent experiments). **i, j** Typical flow cytometry plots (**i**) and quantitative data (**j**) of activated CD8[+] T cells following co-culture with the aforementioned BMDCs (*n* = 3 independent experiments). **k** Representative images showing the process of activated CD8[+] T cells attacking EO771-OVA cells labeled with CellTracker™ Green CMFDA Dye. The T cells were activated by the BMDCs co-cultured with EO771-OVA cells that received CPT&siR CRNPs+C treatment. The experiment was repeated three times independently with similar results. **l, m** Typical flow cytometry plots (**l**) and quantitative data (**m**) of dead EO771-OVA cells after T cell attacking. T cells were activated by BMDCs cocultured with EO771-OVA cells with different formulations (*n* = 3 independent experiments). Statistical analyses were done using one-way ANOVA with Tukey's multiple comparisons and correction. Data are presented as mean ± SD (**a, d, e, h, j, m**). Source data are provided as a Source Data file.

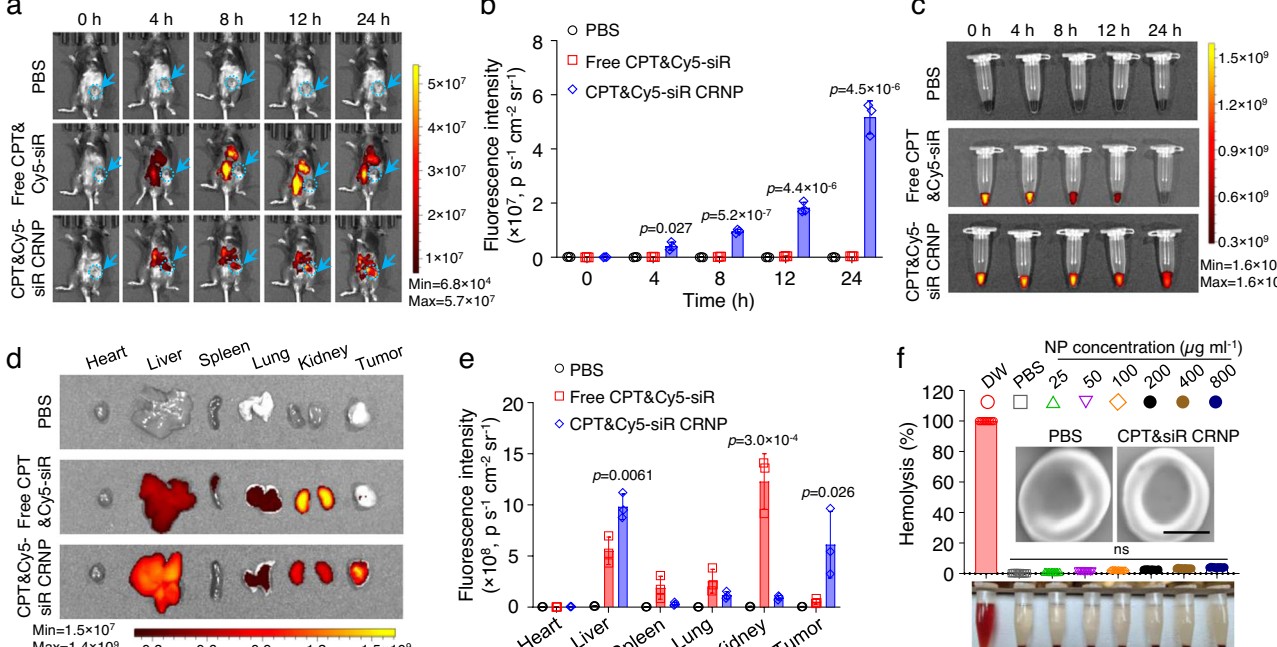

**Fig. 4 | CRNPs show high accumulation in orthotopic breast tumors in vivo and excellent blood compatibility. a, b** Whole animal images (**a**) showing the in vivo distribution of CPT&Cy5-siR CRNPs and the corresponding quantitative data (**b**) of the fluorescence intensity of CPT&Cy5-siR CRNPs in EO771 tumors from IVIS imaging at different time points after intravenous injection of the CRNPs (*n* = 3 mice). PBS and free CPT&Cy5-siR were investigated for comparison. The blue arrows and dashed circles indicate the locations of tumors. **c** Representative images acquired by IVIS imaging showing the Cy5 fluorescence intensity in blood drawn at various time points from mice after injection with PBS, free CPT&Cy5-siR, and CPT&Cy5-siR CRNPs (at the same Cy5-siR dose as free CPT&Cy5-siR) (*n* = 3 mice). **d, e** Typical IVIS images (**d**) and the corresponding quantitative data (**e**) showing the Cy5 fluorescence in different organs including tumors collected from mice injected via tail vein

with PBS, free CPT&Cy5-siR, and CPT&Cy5-siR CRNPs and sacrificed at 24 h after the injections (*n* = 3 mice). **f** Hemolysis assay of CPT&siR CRNPs at different concentrations ranging over 25-800 μg ml⁻¹ (*n* = 3 independent experiments). Deionized water (DW) and PBS were used as the positive (~100% hemolysis) and negative (~0% hemolysis) controls, respectively. Insets are the representative SEM images of red blood cells incubated with PBS and CPT&siR CRNPs at a concentration of 800 μg ml⁻¹. Scale bar: 2 μm. ns: no significance. Statistical analyses were done using one-way ANOVA with Tukey's multiple comparisons and correction. The experiments for **c, d** were repeated three times independently (*n* = 3 mice) with similar results. Data are presented as mean ± SD (**b, e, f**). Source data are provided as a Source Data file.

percentage of cancer cell death (80.5%) than that (61.4%) due to the additive anticancer effect of CPT CRNPs alone and siR CRNPs alone. These in vitro results collectively show that ICIE holds great potential for activating T cells and enhancing their tumor-killing functionality.

### CRNPs accumulate in tumor and exhibit negligible hemolysis

To explore the in vivo distribution of CRNPs, an orthotopic mouse breast tumor model was created by injecting $1 \times 10^6$ EO771 cells per mouse into the left abdominal mammary fat pad of C57BL/6 mice to grow tumors (one per mouse) into approximately 100 mm³ (at around

10 days). After intravenous injection of various formulations (CPT&Cy5-siR CRNPs, free CPT&Cy5-siR, and PBS) into the mice bearing EO771 tumors, Cy5 fluorescence in the mice was studied by whole animal imaging. As shown in Fig. 4a, b, Cy5 fluorescence is evidently observable in the tumor areas (indicated by blue arrows) at 4, 8, 12, and 24 h only for the group of CPT&Cy5-siR CRNPs. At each time point, blood was drawn from the mice for testing Cy5 fluorescence in circulation. Little Cy5 fluorescence signal is observable/detectable (decreased by one order of magnitude) after 8-12 h for the free CPT&Cy5-siR group (Fig. 4c, Supplementary Fig. 11). In contrast, for the CPT&Cy5-siR CRNPs group, the Cy5 fluorescence is evident even at

24 h, showing encapsulation of the Cy5-siR in the CRNPs can prolong its time in blood circulation because the nanoparticle can protect the siR from degradation (Supplementary Fig. 2e). The fluorescence of Cy5 in different organs was further examined after sacrificing the mice at 24 h. Mice treated with free CPT&Cy5-siR show minimal Cy5 fluorescence in tumors, but high Cy5 fluorescence in kidneys and liver and moderate-low fluorescence in the lung and spleen (Fig. 4d, e). In contrast, the Cy5 fluorescence is significantly reduced in kidneys, and significantly increased in liver for the CPT&Cy5-siR CRNPs group (which is not unusual for nanoparticles for drug delivery[55]), compared to the free CPT&Cy5-siR group. Most importantly, the Cy5 fluorescence in tumors is significantly higher for the CPT&Cy5-siR CRNPs group than the free CPT&Cy5-siR group, indicating the tumor accumulating/targeting capability of the CRNPs. Although CPT&Cy5-siR CRNPs show relatively higher accumulation in liver than free CPT&Cy5-siR, drug release from the CRNPs in liver where no cryosurgery will be applied, would be minimal to minimize any potential toxicity of the two encapsulated agents before they are cleared out of the body via the liver.

Since the CRNPs need to go through long blood circulation after intravenous (i.v.) injection, their blood biocompatibility was further examined with the hemolysis test. As shown in Fig. 4f, CPT&siR CRNPs of 25–800 μg ml$^{-1}$ (in PBS) induce a negligible percentage of hemolysis comparable to PBS while DW causes nearly complete hemolysis. Furthermore, compared to PBS, CPT&siR CRNPs (800 μg ml$^{-1}$) do not cause any morphological changes to the red blood cells (SEM images, Fig. 4f) after incubation with the whole blood at 37 °C for 24 h. These data show that no evident blood incompatibility is observable for the CRNPs.

### ICIE turns the TME from immunologically "cold" into "hot" in primary tumors

For these studies, C57BLC/6 mice were inoculated with EO771 cells ($1 \times 10^6$ cells per gland) in the fat pad of both the left and right abdominal mammary glands, to create orthotopic breast tumors. After 10 days of tumor establishment, various formulations (PBS, free CPT&siR, CPT CRNPs, siR CRNPs, and CPT&siR CRNPs) were injected via the tail vein every three days for a total of nine injections either without (Fig. 5a) or with (Fig. 5b) cryosurgery. For the groups with cryosurgery, it was done on the left (in terms of mice) tumors once (blue arrow, Fig. 5b) at 8 h after the first injection of the formulations by gently pressing the cryoprobe against the tumor at its central location (Fig. 5c). Therefore, the left tumors are called primary ones with cryosurgery while the right tumors without cryosurgery are called distant ones for all groups with cryosurgery. It is worth noting that the treatment with cryosurgery being conducted only once and the formulations being injected multiple times, is similar to how frequent cryosurgery (usually once) and chemotherapy (almost always multiple times) are conducted in the clinic. To monitor the temperature during the 10 min of cryosurgery, two K-type thermocouples (TCs) were used by gently pressing them on the surface of the primary tumor (Fig. 5c): one for measuring the temperature at the tumor boundary (TC1) and the other (TC2) for monitoring the temperature at the central location of the tumor next to the cryoprobe. During cryosurgery, the temperature at tumor boundary and central location was kept at approximately −4 and −20 °C for 10 min, respectively (Fig. 5d). This was achieved by turning the cryoprobe on and off intermittently to maintain the iceball roughly within the tumor boundary during the cryosurgery procedure. Moreover, a FLIR infrared camera was used to monitor the temperature in the entire primary tumor area (indicated by dashed circles, Fig. 5e) during cryosurgery. Typical temperature contours in the tumor area together with the temperature on the tumor boundary at 0 s (before cryosurgery), 300 s (during cryosurgery), and 630 s (after cryosurgery and warming up) are also given in Fig. 5e. Typical image showing frozen tumor iceball formation is also

captured at 300 s during cryosurgery (for which the cryoprobe and TCs are temporarily removed).

After the aforementioned treatments, their impact on the TME in the primary tumors was studied. First, the PD-L1 silencing effect of siR in the primary (for mice with cryosurgery) and left (for mice without cryosurgery) tumors was determined by western blot. Both ICIE (i.e., CPT&siR CRNPs with cryosurgery) and siR CRNPs with cryosurgery treatments significantly decrease the PD-L1 expression in tumors when compared with CPT CRNPs with cryosurgery or PBS with cryosurgery (Supplementary Fig. 12a). In the absence of cryosurgery, the silencing effect is not evident for all the formulations including siR CRNPs or CPT&siR CRNPs (Supplementary Fig. 12b). Second, the expression of ICD markers (i.e., DAMPs including HMGB1, CRT, HSP-70, and HSP-90) is markedly increased in the primary tumors from the groups with cryosurgery (Fig. 5f), compared to their counterparts without cryosurgery (Supplementary Fig. 13). Notably, The ICIE treatment stimulates higher expression of all the ICD markers in the primary tumors than all the other treatments with or without cryosurgery. Third, compared to the percentage of tumor-associated macrophages (TAMs) in left tumors (corresponding to primary tumors in mice with cryosurgery) of mice without cryosurgery, cryosurgery not only decreases the population of pro-tumorigenic M2 TAMs (F4/80$^+$CD206$^+$CD86$^-$) but also elevates the percentage of anti-tumorigenic M1 TAMs (F4/80$^+$CD206$^-$CD86$^+$) in all treated groups (Supplementary Fig. 14a–d). As a result, a marked increase in the M1/M2 ratios is observable in mice with cryosurgery (Fig. 5g), with the highest M1/M2 ratio of $4.0 \pm 0.5$ in mice receiving the ICIE treatment. Fourth, cryosurgery evidently attenuates the frequency of both monocytic myeloid-derived suppressor cells (M-MDSCs, CD11b$^+$Ly6C$^+$Ly6G$^-$) and polymorphonuclear myeloid-derived suppressor cells (PMN-MDSCs, CD11b$^+$ Ly6C$^-$Ly6G$^+$) that perform their immunosuppressive activities within the TME (Fig. 5h, i and Supplementary Fig. 15a–d). The ICIE treatment decreases the percentage of M-MDSCs and PMN-MDSCs to $8.8 \pm 1.0\%$ and $16.1 \pm 2.1\%$, respectively, which is significantly lower than that for its counterpart (CPT&siR CRNPs) with no cryosurgery ($18.9 \pm 1.4\%$ for M-MDSCs and $35.4 \pm 3.5\%$ for PMN-MDSCs). Furthermore, in the primary tumor, the percentage of regulatory T (Treg, CD4$^+$Foxp3$^+$) cells is greatly reduced by the ICIE treatment (Fig. 5j and Supplementary Fig. 15e, f). The PBS control group (without cryosurgery) had a high percentage of Treg ($42.9 \pm 3.0\%$), indicating an immunosuppressive TME[56]. The cryosurgery treatment can significantly reduce the percentage of Treg cells for all the formulations. Importantly, the ICIE treatment results in the most reduction of the Treg cell percentage to only $1.6 \pm 0.7\%$ (Fig. 5j and Supplementary Fig. 15e). Collectively, these data support that the ICIE treatment induces ICD; produces tumor antigens; and decreases the frequency of immunosuppressive M2, MDSCs, and Tregs; to reverse the immunologically cold TME. This is attributed to the cryosurgery-induced ICD reinforced by the cold-triggered rapid release of both the chemotherapy drug irinotecan (CPT) that inhibits DNA synthesis[49,57], and the siRNA (siR) that silences the overexpression of T cell inhibitory ligand PD-L1 on cancer cells to remove the immune checkpoint that inhibits the activation of CD8$^+$ T cells[53,54]. In addition, the frostbite effect of cryosurgery might kill the M2, MDSCs, and Tregs in the immunosuppressive TME, to reduce their frequency in the tumor after cryosurgery. As a result, ICIE is highly effective to turn the immunologically cold TME into a hot one.

### ICIE boosts the antitumor immune responses in primary tumors

To assess if the aforementioned modulation of TME by ICIE boosts the antitumor immune responses, we analyzed the population of DCs and

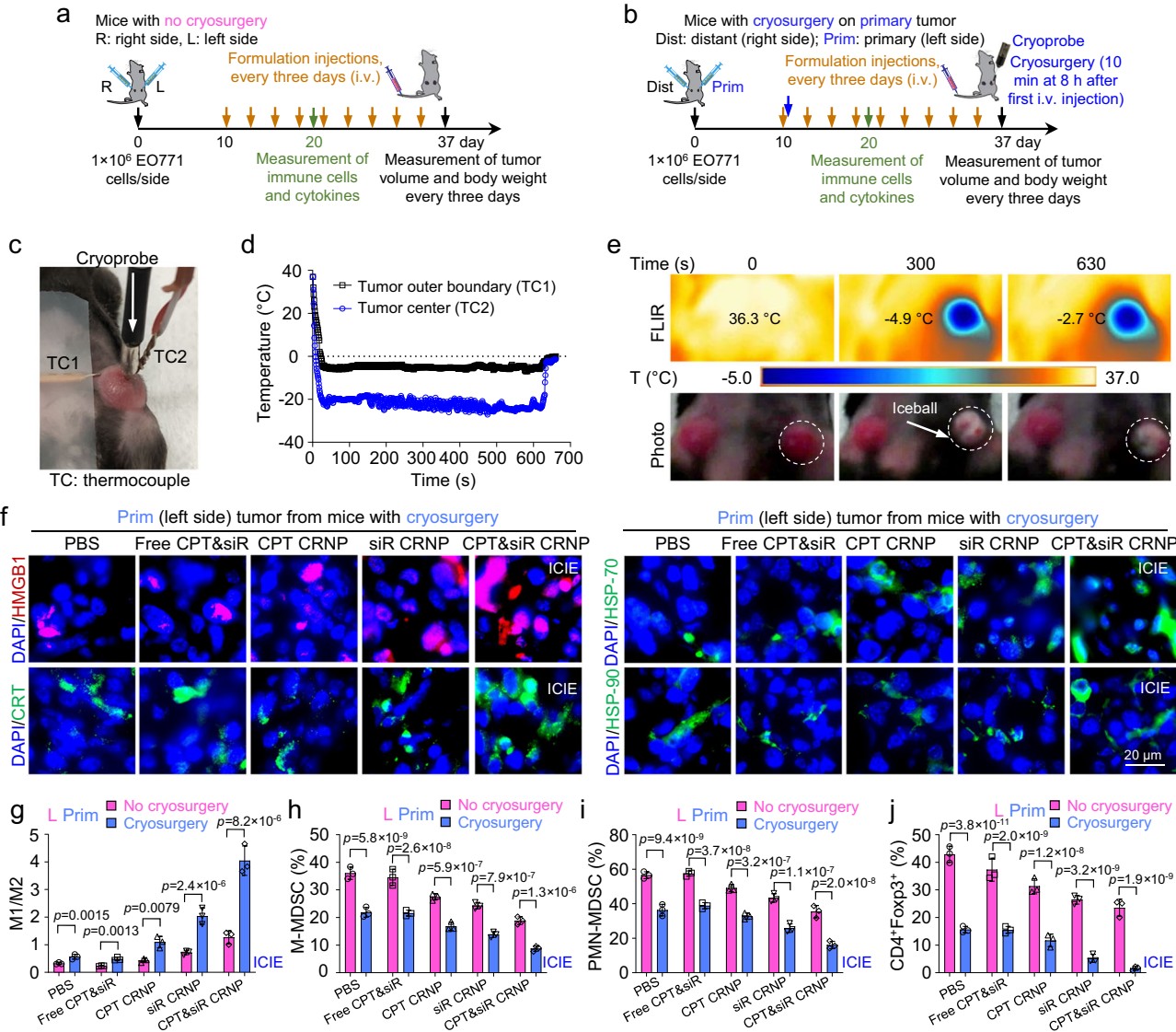

**Fig. 5 | ICIE turns immunologically "cold" TME into a "hot" one. a** A schematic illustration of the experimental design for mice receiving only intravenous injection of various formulations every 3 days with no cryosurgery. For these mice, the tumors on the left and right sides are indicated as L and R tumors, respectively. **b** A schematic illustration of the in vivo experimental design for mice receiving both intravenous injection of various formulations and cryosurgery done only on the primary (Prim) tumors on the left side. The tumors that undergo no freezing on the right side of these mice are indicated as distant (Dist) tumors. Cryosurgery was performed at 8 h after the first injection of the formulations. **c** A typical photograph of the tumor area of a mouse during cryosurgery. Thermocouple 1 (TC1) and TC2 are for monitoring the temperature in tumor outer boundary and center (next to the cryoprobe), respectively. **d** Typical thermal histories recorded at the tumor boundary and center during the cryosurgical procedure. **e** Representative FLIR infrared images and photographs of mice showing the tumor temperature and appearance at three different time points before (0 s), during (300 s), and after

(630 s) cryosurgery. The dashed circles indicate the tumor areas. **f** Representative immunofluorescence images showing the expression of HMGB1, CRT, HSP-70, and HSP-90 in primary tumors treated by the indicated formulations in combination with cryosurgery. **g–j** Quantitative flow cytometry data on the ratio of tumor associated macrophages M1 (F4/80$^+$CD206$^-$CD86$^+$) to M2 (F4/80$^+$CD206$^+$CD86$^-$) (**g**), percentage of monocytic myeloid-derived suppressor cells (M-MDSCs, CD11b$^+$Ly6C$^+$Ly6G$^-$, **h**), percentage of polymorphonuclear myeloid-derived suppressor cells (PMN-MDSCs, CD11b$^+$Ly6C$^-$Ly6G$^+$, **i**), and percentage of regulatory T (Treg) cells (CD4$^+$Foxp3$^+$, **j**) in primary (Prim) tumors of mice with both cryosurgery and injection of one of the formulations and in L (left side) tumors of mice injected with the different formations only and with no cryosurgery (*n* = 3 mice). Statistical analyses were done using two-way ANOVA with Sidak's post-test and correction for multiple comparisons. The experiments for **e** and **f** were repeated three times independently (*n* = 3 mice) with similar results. Data are presented as mean ± SD (**g–j**). Source data are provided as a Source Data file.

CD8$^+$ T cells in the mice. As shown in Fig. 6a and Supplementary Fig. 16a, b, cryosurgery promotes maturation of DCs in the inguinal lymph nodes, leading to significantly increased percentages of mature DCs (CD11c$^+$CD86$^+$) for all the groups with cryosurgery. In particular, the ICIE treatment (i.e., CPT&siR CRNPs with cryosurgery) triggers the highest percentage of DC maturation among all the treatments (with or without cryosurgery). Furthermore, in the primary tumors where the cryosurgery is applied, the augmented DC maturation enables significantly higher accumulation of tumor-infiltrating lymphocytes

(Fig. 6b and Supplementary Fig. 16c). Importantly, the percentage of tumor-infiltrating CD8$^+$ T cells for the ICIE group is the highest (23.8 ± 0.6% out of all T cells) among all groups, with a highest CD8$^+$/Treg ratio of 17.6 ± 10.3 (Fig. 6c) that is 117 times of the CD8$^+$/Treg ratio in the L tumor with the PBS treatment alone. This indicates the ICIE treatment turns the immunologically "cold" TME into a "hot" one. In contrast, the treatment of CPT&siR CRNPs without cryosurgery only results in 9.5 ± 0.8% of tumor infiltrating CD8$^+$ T cells (Supplementary Fig. 16d) and a CD8$^+$/Treg ratio of 0.4 (Fig. 6c), indicating an

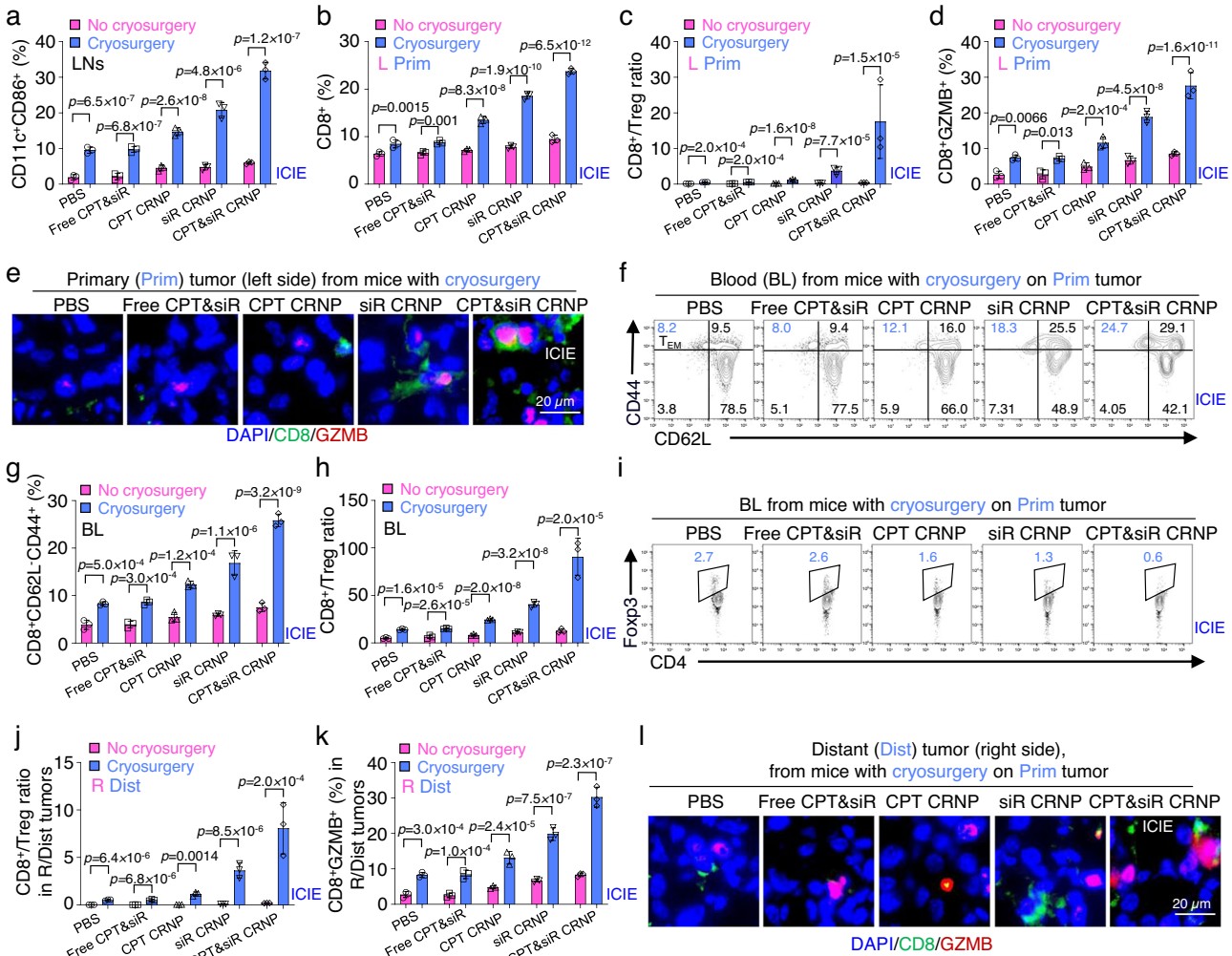

**Fig. 6 | ICIE stimulates antitumor immune responses in primary tumors and activates the memory immune response against distant tumors.**
**a–d** Quantitative flow cytometry data of matured DCs (CD11c⁺CD86⁺) in inguinal lymph nodes (LNs, **a**), and percentage of infiltrated CD8⁺ T cells (**b**), ratios of CD8⁺/Treg cells (**c**), and percentage of cytotoxic T cells (CTLs, CD8⁺GZMB⁺, **d**) in primary (Prim) tumors of mice with cryosurgery and L (left side) tumors of mice with no cryosurgery (*n* = 3 mice). **e** Representative immunofluorescence images of infiltrated CTLs in primary (Prim) tumors from mice with cryosurgery (*n* = 3 mice). **f** Representative flow cytometry plots showing the percentage of effector memory T cell (T$_{EM}$, CD3⁺CD8⁺CD44⁺CD62L⁻) in blood (BL) collected from mice with cryosurgery and injection of one of the various formulations. **g** Quantitative flow cytometry data of T$_{EM}$ in blood of mice with/without cryosurgery and injection of one of the different formulations (*n* = 3 mice). **h** CD8⁺/Treg ratios in the blood of mice with/without cryosurgery and injection of one of the different formulations (*n* = 3 mice). **i** Representative flow cytometry plots of Treg cells in blood collected from mice with cryosurgery and injection of one of the different formulations. **j** Quantitative flow cytometry data of CD8⁺/Treg ratios in distant (Dist) tumors of mice with cryosurgery and R (right side) tumors of mice without cryosurgery (*n* = 3 mice). All mice were injected with one of the indicated formulations. **k** Quantitative flow cytometry data of the percentage of infiltrated CTLs in Dist tumors of mice with cryosurgery and R (right side) tumors of mice without cryosurgery (*n* = 3 mice). All mice were injected with one of the indicated formulations. **l** Representative immunofluorescence images of infiltrated CTLs in Dist tumors from mice with cryosurgery on Prim tumors and injection of one of the different formulations (*n* = 3 mice). Statistical analyses were done using two-way ANOVA with Sidak's post-test and correction for multiple comparisons. The experiments for **e** and **l** were repeated three times independently (*n* = 3 mice) with similar results. Data are presented as mean ± SD (**a–d**, **g**, **h**, **j**, **k**). Source data are provided as a Source Data file.

immunologically "cold" TME. Similarly, the frequency of CD8⁺GZMB⁺ CTLs is the highest (27.6 ± 3.7% out of all CD8+ T cells) for the ICIE group (Fig. 6d & Supplementary Fig. 16e, f), which is confirmed by immunofluorescence staining images (Fig. 6e and Supplementary Fig. 16g) showing the distribution of CD8⁺GZMB⁺ cells in tumor sections. It is worth noting that unlike hyperthermic therapy for which permanent vascular stasis occurs due to heating, there is a temporary (a few hours) reperfusion of the tumor immediately after thawing a frozen tumor iceball[27–29], which may allow efficient immune cell infiltration into the tumor after cryosurgery to boost the antitumor responses. Collectively, these data support that the ICIE treatment may activate a strong antitumor immunity to kill cancer cells through a high level of GZMB secreted by CTLs.

## ICIE generates both short-term and long-term memory immune effects

To explore whether the activation of immune responses in primary tumor generates memory immune effects, the amount of effector memory T (T$_{EM}$, CD8⁺CD44⁺CD62L⁻) cells was measured first. The T$_{EM}$ cells circulate in the blood and are ready to exert direct and rapid cytotoxicity against any existing tumor cells[58,59]. Consistent with the improved antitumor immune responses in primary tumors, ICIE treatment significantly augments the proportion of T$_{EM}$ cells to 25.8 ± 1.3% from 3.9 ± 1.0% for PBS without cryosurgery (Fig. 6f and Supplementary Fig. 17a). Also, cryosurgery significantly increases the T$_{EM}$ cell proportion for all the formulations, compared to their counterparts without cryosurgery (Fig. 6g). In addition, ICIE promotes the

percentage of central memory T cells ($T_{CM}$, $CD8^+CD44^+CD62L^+$) in spleen to $45.3 \pm 1.4\%$ from $19.5 \pm 0.7\%$ for PBS without cryosurgery (Supplementary Fig. 17b–d), and mice received combinational treatments of cryosurgery and aforementioned formulations show a significantly higher frequency of $T_{CM}$ than those with no cryosurgery (Supplementary Fig. 17d), suggesting the capability of generating long-term protection against tumor burdens, metastasis, or relapse.

## ICIE boosts the antitumor immune responses in blood and distant tumors

As $CD8^+$ T cells activated by the hot TME in primary tumor need to go with blood to enter distant tumors, we studied the $CD8^+$ T cells together with Treg cells in blood first. The ratio of $CD8^+$ to Treg cells ($CD8^+$/Treg), an important value that has been used to predict the antitumor immune response and treatment outcome[60,61], was therefore determined using whole blood samples. ICIE dramatically increases the $CD8^+$/Treg ratio to $90.7 \pm 19.3$ from $5.0 \pm 0.8$ for control treatment of PBS without cryosurgery (Fig. 6h, i and Supplementary Fig. 17e–g). Again, for all the formulations, cryosurgery significantly increases the $CD8^+$/Treg ratio in blood. Importantly, the ratio of $CD8^+$ to Treg cells for the ICE treatment is higher than that for all the other treatments including CPT CRNPs, siR CRNPs, and free CPT&siR either with or without cryosurgery. Notably, the subpopulation of Treg cells in blood is almost unidentifiable following the ICIE (i.e., CPT&siR CRNPs with cryosurgery) treatment (Fig. 6i), while the Treg subpopulation is evident for the treatment of CPT&siR CRNPs without cryosurgery ($2.8 \pm 0.5\%$, Supplementary Fig. 17g). These results suggest that the ICIE treatment in primary tumors is very promising in boosting the antitumor immune responses to distant tumors.

Indeed, the ICIE treatment (i.e., CPT&siR CRNPs with cryosurgery) on the primary tumors increases the $CD8^+$/Treg ratio to $8.1 \pm 2.7$ in distant (Dist) tumors (without freezing) in the same mice, which is 376.2-folds higher than that for the right-side (R) tumors of the PBS control from mice with no cryosurgery (Fig. 6j and Supplementary Fig. 18a–d). In contrast, a significantly lower $CD8^+$/Treg ratio (0.2) is observed in the distant tumors from mice treated with CPT&siR CRNPs without cryosurgery. Although it is true that cryosurgery enhances $CD8^+$/Treg ratios for all the formulations studied (Fig. 6j), the increase in the ratio is the most evident for the two formulations with siR. This suggests not only the cold-triggered drug/gene release but also the cold-triggered endo/lysosomal escape, in addition to the freezing-induced immune responses, is crucial to render the ICIE treatment with the potent antitumor immune responses against both primary and distant tumors. Moreover, ICIE elevates the percentage of $CD8^+GZMB^+$ T cells in distant tumors to $30.2 \pm 2.8\%$ (out of all $CD8^+$ T cells, Fig. 6k and Supplementary Fig. 18e), which is 3.6-folds higher than that ($8.3 \pm 0.5\%$) for its non-cryosurgical counterpart (Supplementary Fig. 18f). Immunofluorescence staining of the distant tumors further confirms the aforementioned observations, showing more accumulation of $CD8^+GZMB^+$ T cells in the ICIE than other groups (Fig. 6l and Supplementary Fig. 18g). Taken together, these data suggest that ICIE is promising for destroying not only primary but also distant tumors, which is further confirmed by tumor growth studies as detailed below.

## ICIE effectively inhibits the growth of both primary and distant tumors

To evaluate the therapeutic efficacy of the aforementioned antitumor immune responses, the dual orthotopic tumor model formed by injecting EO771 cells into both the left-side (L tumors for mice with no cryosurgery and primary tumors for mice with cryosurgery) and right-side (R tumors for mice with no cryosurgery and distant tumors for mice with cryosurgery done on the primary tumors) abdominal mammary fat pads of C57BL/6 mice as schematically illustrated in Fig. 5a, b, was continuously observed for 37 days. As shown in Fig. 7a, b and Supplementary Fig. 19a–d, cryosurgery evidently decreases the growth of both primary and distant tumors from mice with cryosurgery on their primary tumors for all the formulations, compared to that of the L and R tumors from mice with no cryosurgery. In particular, the ICIE treatment (i.e., CPT&siR CRNPs with cryosurgery) is the most effective in either inhibiting the tumor growth or inducing complete tumor eradication among all the treatments (with or without cryosurgery). In addition, in both the primary (left) and distant (right) tumors, the treatment of CPT&siR CRNPs results in lower tumor volumes than other formulations (free CPT&siR, CPT CRNPs and siR CRNPs) in the presence and absence of cryosurgery (Supplementary Fig. 19a–d). The therapeutic efficacy was further confirmed by tumor images and tumor weights obtained at the final day (day 37) of the in vivo study. Again, for all the formulations, cryosurgery evidently reduces tumor sizes (Supplementary Fig. 20a–c) and the reduction is significant for all the formulations according to the data of tumor weights (Fig. 7c, d). Also, either with or without cryosurgery, more inhibition of tumor growth and less tumor weight are observable for the formulation of CPT&siR CRNPs than all the other formulations including free CPT&siR, CPT CRNPs and siR CRNPs.

To further investigate the ICIE-induced tumor destruction, we sectioned the primary tumors from mice with cryosurgery and L tumor from mice with no cryosurgery and stained them for Ki-67 (representing tumor cell proliferation) and CD31 (representing angiogenesis). Similar to the tumor inhibition results, cryosurgery evidently downregulates the expression of both Ki-67 (Supplementary Fig. 21a) and CD31 (Supplementary Fig. 21b) in tumors treated with all the formulations, with the ICIE treatment results in the most reduction. This indicates the strongest inhibition of both tumor cell proliferation and tumor vascularization with the ICIE treatment. The antitumor effect was further investigated using hematoxylin & eosin (H&E) staining, which shows that necrosis is the most extensive in both primary and distant tumors from the ICIE group among all the other groups either with or without cryosurgery (Fig. 7e).

It is worth noting that no significant change of body weight was observed for mice treated with all formulations either with or without cryosurgery (Supplementary Fig. 22a, b). Also, H&E staining analyses show no significant damages in the major organs (heart, liver, spleen, lung, and kidney) harvested from mice sacrificed at the end of the in vivo study for all the treatments with therapeutic agents and/or cryosurgery, when compared to that of the PBS control group (Supplementary Figs. 23 and 24), suggesting the good biosafety of CRNPs. Furthermore, no significant changes in the alanine aminotransferase (ALT) and aspartate aminotransferase (AST) levels can be detected in the blood of mice after treatment with CPT CRNPs, siR CRNPs, or CPT&siR CRNPs when compared to PBS group (Supplementary Fig. 25a, b), confirming that the CRNPs cause negligible damage to the liver although they do have high accumulation in the liver after i.v. injection (Fig. 4d, e). Again, the minimal release of the encapsulated drugs from the CRNPs in the liver where no cryosurgery is performed, contributes to the negligible liver toxicity of the three different CRNPs. Collectively, these results indicate that ICIE generates an antitumor immune response to effectively kill both primary and distant tumors with no evident systemic toxicity.

## ICIE generates long-term antitumor memory immune response against metastatic tumors

To further evaluate the generation of long-term antitumor memory immune response by ICIE for preventing lung metastasis, Balb/c mice with 4T1 orthotopic tumors (OTs) were applied with different treatments including ICIE (Fig. 8a). After priming for 10 days, a lung metastatic tumor model was formed by i.v. injection of 4T1 cancer cells ($1 \times 10^5$ cells/mouse) and the mice were further injected with the various formulations till death or termination of experiments (Fig. 8a). The results show that ICIE can not only significantly inhibit the growth of orthotopic 4T1 tumor and decrease the orthotopic tumor weight to

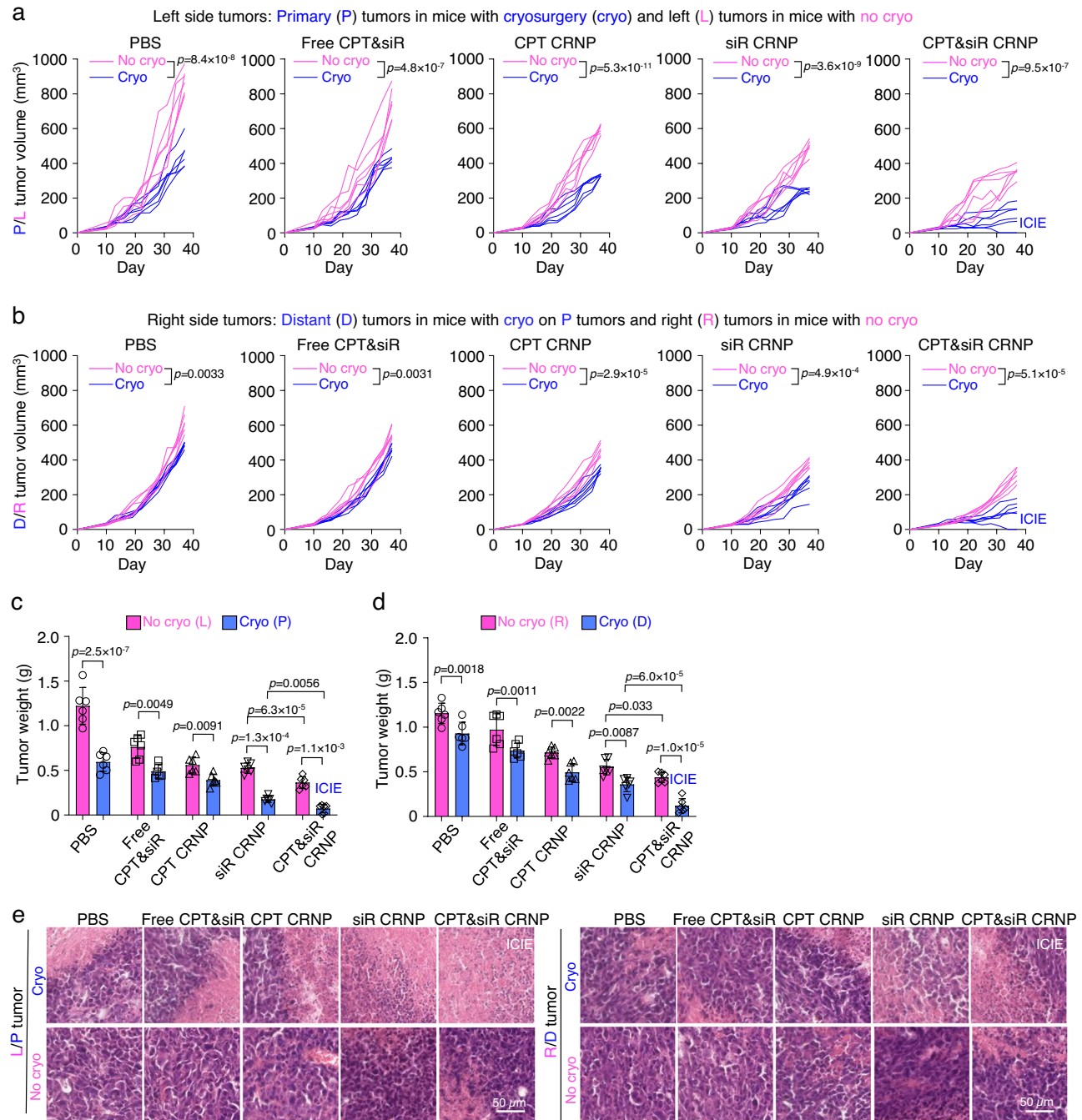

**Fig. 7 | ICIE effectively inhibits tumor growth in both primary and distant tumors with no evident systemic toxicity. a** Tumor growth curve of left-side tumors including primary (P) tumors from mice with cryosurgery and left (L) tumors from mice without cryosurgery (*n* = 6 mice). The mice were also injected with one of the formulations including: PBS, free CPT&siR, CPT CRNPs, siR CRNPs, or CPT&siR CRNPs. **b** Tumor growth curve of right-side tumors including distant (D) tumors from mice with cryosurgery on P tumors and right (R) tumors from mice without cryosurgery (*n* = 6 mice). The mice were also injected with one of the indicated formulations. **c, d** Weight of left-side (L versus P, **c**) and right-side (R versus D, **d**) tumors (*n* = 6 mice) obtained after sacrificing the mice at the end of the study, showing effective tumor destruction by the ICIE treatment. **e** Representative hematoxylin and eosin (H&E) staining images of left-side (L versus P) and right-side (R versus D) tumors, showing extensive necrosis in both the primary and distant tumors from mice with the ICIE treatment. The experiments were repeated three times independently (*n* = 3 mice) with similar results. Statistical analyses were done using two-way ANOVA with Sidak's post-test and correction for multiple comparisons. Data are presented as mean ± SD (**c, d**). Source data are provided as a Source Data file.

2.6-fold and 2.9-fold less than that of CPT&siR CRNPs without cryosurgery on orthotopic tumor and PBS group with cryosurgery on orthotopic tumor, respectively (Supplementary Fig. 26a), but also reduce the metastatic tumor burden in lungs, resulting in the lowest weight of lungs (0.21 ± 0.03 g, which is close to the weight of normal lung[62]) amongst all treatment groups (Fig. 8b). As shown in the representative images of lungs collected at the end of the study (Fig. 8c), ICIE greatly reduces the formation of metastatic foci in the lungs: the number of metastatic foci for the ICIE are 3.2-fold and 3.6-fold less than that of CPT&siR CRNPs without cryosurgery on orthotopic tumor and PBS group with cryosurgery on orthotopic tumor, respectively (Fig. 8d). This is further confirmed by the H&E staining

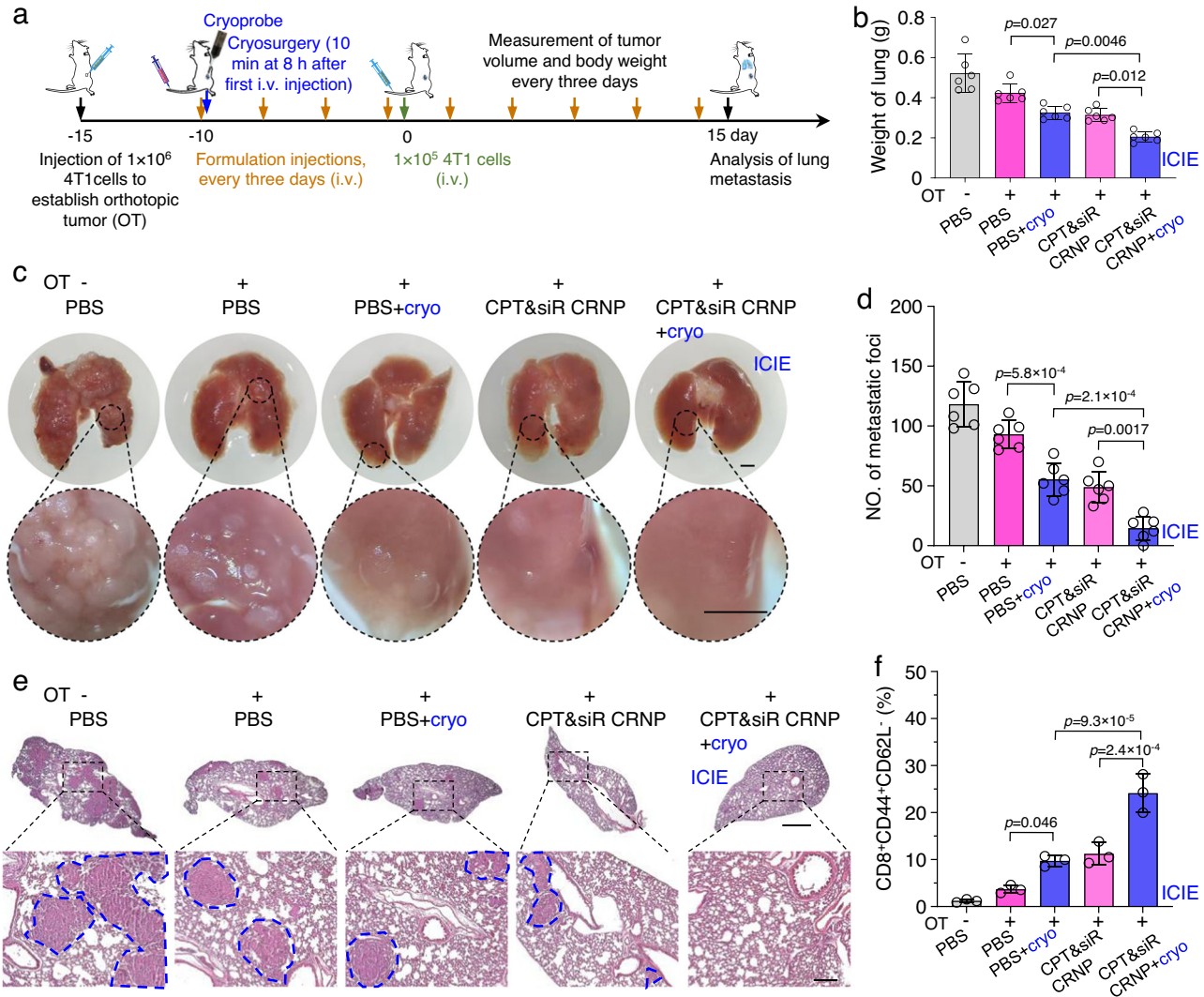

**Fig. 8 | ICIE generates long-term antitumor memory immune response and inhibits metastasis in the lung. a** A schematic illustration of the in vivo experimental design. Mice received an intravenous injection of various formulations every 3 days. Cryosurgery was performed on the orthotopic tumor (OT) at 8 h after the first injection of the various formulations. The lung metastasis model was induced by intravenous injection of 4T1 cancer cells on day 10 after the first injection of the different formulations. **b** Weight of lungs collected from mice at the end of the study, suggesting inhibition of lung metastasis by ICIE ($n = 6$ mice). **c** Representative photographs of lung tissues isolated at the end of the study, showing inhibition of lung metastasis by the ICIE treatment. Scale bar: 1 mm. **d** Quantification of lung metastasis nodes per mouse from mice at the end of the study ($n = 6$ mice). **e** Representative H&E staining images of lung tissues collected at the end of the study, showing the decreased formation of metastasis in the lungs of mice with the ICIE treatment. Scale bar: 1 mm for low-magnification images and 200 μm for zoom-in images. The areas circled by blue dashed lines are metastases in the lungs. The experiments were repeated three times independently ($n = 3$ mice) with similar results. **f** The percentage of effector memory T cells ($T_{EM}$, $CD3^+CD8^+CD44^+CD62L^-$) in blood collected from mice with the various treatments ($n = 3$ mice). Statistical analyses were performed using one-way ANOVA with Tukey's multiple comparisons test and correction. Data are presented as mean ± SD (**b**, **d**, **f**). Source data are provided as a Source Data file.

data of lungs (Fig. 8e), showing the least metastasis and most alveolar areas after ICIE treatment when compared to all other treatment groups. The strong inhibition of lung metastasis can be ascribed to the generation of the effector memory T ($T_{EM}$, $CD8^+CD44^+CD62L^-$) cells: ICIE significantly increases the effector memory T cells by 2.2 and 2.5 times compared with the CPT&siR CRNPs without cryosurgery on orthotopic tumor and PBS group with cryosurgery on orthotopic tumor, respectively (Fig. 8f and Supplementary Fig. 26b). As a result, the survival rate is significantly improved to 73% for mice with the ICIE treatment, while the survival rate of mice for the treatments of CPT&siR CRNPs without cryosurgery on orthotopic tumor and PBS group with cryosurgery on orthotopic tumor is 50% and 47%, respectively (Supplementary Fig. 27). These results support that ICIE can be applied to generate a durable long-term antitumor memory immune

response to inhibit tumor metastasis and prolong overall animal survival.

## Discussion

ICIE, a cryo-immunotherapeutic strategy that uses cryosurgery and cold-responsive nanomaterials loaded with CPT and PD-L1 silencing siRNA, is developed in this study. Cryosurgery induces ICD, leading to the expression of DAMPs like HMGB1, CRT, HSP-70, and HSP-90 and maturation of DCs, which activates $CD8^+$ T cells to attack tumor cells. When cooled below their LCST during cryosurgery, the cold-responsive nanoparticles rapidly release the encapsulated therapeutic agents to induce ICD and downregulate PD-L1 expression in breast cancer cells. This further potentiate the cryo-immune responses. As a result, ICIE reverses the immunosuppressive TME, and

generates a robust antitumor and long-term antitumor memory immune response for eradicating both primary and distant as well as metastatic tumors in vivo with no evident systemic toxicity. The ICE strategy, which combines cryosurgery, cold-responsive nanomaterial, and immunotherapy, has the potential to transform the treatment of cancer and its metastasis for which conventional methods of cancer cryosurgery, chemotherapy, and immunotherapy alone or their dual combinations have fallen short.

## Methods

### Animals and ethics statement

Female C57BL/6 wildtype (4-week-old) and C57BL/6 OT-I (4-week-old) mice were purchased from Charles River (Wilmington, MA, USA), and The Jackson Laboratory (Bar Harbor, ME, USA), respectively. Female Balb/cJ mice (4-week-old) were ordered from The Jackson Laboratory. All the mice (three to five mice per cage) were housed in standard, infection-free housing room, with 12 h light:12 h dark cycles, stable temperature at ~23 °C, and stable humidity of ~40% in the vivarium at the University of Maryland, College Park. All procedures and animal cares were in accordance with the animal protocol (#R-May-18-24) approved by the Institutional Animal Care and Use Committee (IACUC) at the University of Maryland, College Park.

### Materials

Butyl acrylate (BA), dioxane, 2,20-azobis(2-methylpropionitrile) (AIBN), tetrahydrofuran, poly (D, L-lactide-co-glycolide) (PLGA, lactide/glycolide: 75/25, Mw: 4,000-15,000), MISSION® siRNA Fluorescent Universal Negative Control #1 (Cyanine 5), and Pluronic F127 (PF-127), were purchased from Sigma (St. Louis, MO, USA). The phospholipid 1,2-dipalmitoyl-sn-glycero-3-phosphocholine (DPPC) was ordered from Echelon Biosciences (Salt Lake City, UT, USA). N-isopropylacrylamide (NIPAAm), poly (vinyl alcohol) (PVA, Mw: 30,000-70,000 Da), Silencer™ GFP (eGFP) siRNA, CellTracker™ Green CMFDA Dye, and diethyl ether, were purchased from Thermo Fisher scientific (Waltham, MA, USA). Pharmaceutical grade chitosan oligosaccharide (Mw = 1.2 kDa, 95% deacetylation) was purchased from Zhejiang Golden Shell Biochemical Co. Ltd (Zhejiang, China). The chitosan-modified PF-127 (CS-PF-127) was synthesized and purified using our previously reported method[49]. Irinotecan (CPT) was purchased from Selleckchem (Houston, TX, USA). Anti-mouse/rat ki-67 antibody (14-5698-82, clone: SolA15, dilution of 1:200), Alexa Fluor™ 488 goat anti-mouse IgG (A11001, dilution of 1:500), Alexa Fluor™ 488 goat anti-rabbit IgG (A11008, dilution of 1:500), Alexa Fluor™ 568 goat anti-mouse IgG (A11004, dilution of 1:500), Alexa Fluor™ 568 goat anti-rabbit IgG (A11011, dilution of 1:500) was purchased from Invitrogen (Waltham, MA, USA). The PD-L1 silencing siRNA (Pdcd-1L1 siRNA (m), sc-39700) and anti-mouse PD-L1 antibody (sc-518027, clone: D8, dilution of 1:1,000) were ordered from Santa Cruz (Dallas, TX, USA). Anti-rabbit HMGB1 antibody (3935 S, dilution of 1:1,000), anti-rabbit calreticulin antibody (2891 S, dilution of 1:1,000), anti-rabbit HSP-70 antibody (4872 S, dilution of 1:1,000), anti-rabbit HSP-90 antibody (4874 S, dilution of 1:1,000), anti-rabbit GAPDH antibody (2118 S, clone: 14C10, dilution of 1:1,000), horseradish peroxidase (HRP)−anti-rabbit IgG antibody (7074P2, dilution of 1:1,000) and HRP−anti-mouse IgG antibody (7076P2, dilution of 1:1,000) were purchased from Cell Signaling Technology (Danvers, MA, USA). Anti-mouse CD31 antibody (ab28364, dilution of 1:200) was ordered from Abcam (Cambridge, United Kingdom). The antibodies for flow cytometry including Brilliant Violet 650™ anti-mouse CD206 (MMR) (141723, clone: C068C2, dilution of 1:200), Brilliant Violet 510™ anti-mouse/human CD11b (101263, clone: M1/70, dilution of 1:200), Brilliant Violet 605™ anti-mouse CD45 (103155, clone: 30-F11, dilution of 1:200), Brilliant Violet 785™ anti-mouse F4/80 (123141, clone: BM8, dilution of 1:200), Alexa Fluor® 647 anti-mouse Ly-6C (128010, lone: HK1.4, dilution of 1:200), Alexa Fluor 700 anti-mouse

Ly-6G (127622, clone: 1A8, dilution of 1:200), APC anti-mouse CD62L (104412, clone: MEL-14, dilution of 1:200), Alexa Fluor® 647 anti-mouse/rat/human FOXP3 (320014, clone: 150D, dilution of 1:200), PE anti-mouse/human CD44 (103008, clone: IM7, dilution of 1:200), Alexa Fluor® 700 anti-mouse CD3 (100216, clone: 17A2, dilution of 1:200), Pacific Blue™ anti-mouse CD4 (100428, clone: GK1.5, dilution of 1:200), APC/Cyanine7 anti-mouse CD8a (100714, clone: 53-6.7, dilution of 1:200), PE anti-mouse CD11c (117308, clone: N418, dilution of 1:200), FITC anti-mouse CD86 (105006, clone: GL-1, dilution of 1:200), APC anti-mouse H-2Kb bound to SIINFEKL (141606, clone: 25-D1.16, dilution of 1:200), FITC anti-human/mouse granzyme B (GZMB) (515403, clone: GB11, dilution of 1:200), and PE anti-mouse CD274 (124308, clone: MIH7, dilution of 1:200), were purchased from Biolegend (San Diego, CA, USA). All other chemicals were ordered from Sigma unless specifically indicated otherwise.

### Cells

EO771 medullary breast adenocarcinoma cells (CRL-3461) and 4T1 cells (CRL-2539) were purchased from ATCC (Manassas, VA, USA). EO771-OVA cells and GFP⁺ EO771 cells were generated as previously reported[63]. Briefly, to generate the EO771-OVA cell line, EO771 cells were transduced with a lentiviral vector expressing chicken ovalbumin gene, and selected for stable expression colonies via puromycin (1 μg ml⁻¹) resistance and OVA presentation levels. A stable cell colony with >50% of the cells positive for OVA presentation in flow cytometry analysis was selected and propagated for further use. To generate the GFP⁺ EO771 cell line, EO771 cells were transduced with a lentiviral vector expressing eGFP. At 48 h after infection, the cells were selected with puromycin (1 μg ml⁻¹) for 7 days and GFP-positive cells were collected by cell sorting. A stable GFP⁺ cell line was selected and expanded for further use. EO771 cells and 4T1 cells were cultured in Dulbecco's Modified Eagle Medium (DMEM, Gibco, Gaithersburg, MD, USA) and Roswell Park Memorial Institute (RPMI) 1640 Medium (Gibco), respectively, supplemented with 1% penicillin−streptomycin (PS) and 10% fetal bovine serum (FBS) under 5% $CO_2$ at 37 °C in a humidified incubator. Bone marrow dendritic cells (BMDCs) were collected from femurs of 6-week-old C57BL/6 mice and cultured in RPMI medium containing interleukin-4 (IL-4, 10 ng ml⁻¹, Peprotech, Cranbury, NJ, USA) and granulocyte-macrophage colony-stimulating factor (GM-CSF, 20 ng ml⁻¹, Peprotech) for 7 days. To culture the CD8⁺ T cells, single cell suspension was harvested from spleens and inguinal lymph nodes of 6-week-old C57BL/6 OT-I mice) through mechanical dissociation[64], and then purified by CD8a (Ly-2) MicroBeads (Miltenyi Biotec, Bergisch Gladbach, Germany). The resultant CD8⁺ T cells were maintained in RPMI medium supplemented with 10% FBS and 10 ng ml⁻¹ IL-2 (Peprotech) under 5% $CO_2$ at 37 °C in a humidified cell incubator.

### Synthesis and characterization of pNIPAAm-BA

The pNIPAAm-BA polymers with different ratios of NIPAAm to BA were synthesized by controlling the copolymerization of NIPAAm and BA using a reported procedure[65]. Briefly, for the polymerization of the ratio of 81:19 (NIPAAm:BA), 16.2 mmol (1.834 g) of NIPAAm, and 3.8 mmol (0.488 g) of BA were transferred into a flask and dissolved in 10 ml of dioxane. After purging the resultant solution with nitrogen gas for 30 min, 11.61 mg of AIBN dissolved in 5 ml of dioxane was added dropwise into the flask. The flask was then immersed into an oil-bath at 70 °C with the solution in it being stirred at 400 rpm for 12 h under a nitrogen atmosphere. The polymerization was stopped by exposing the flask to air for cooling down to room temperature (~22 °C). Afterwards, the resultant sample was precipitated in excessive amount of cold diethyl ether and filtered through a filter paper (size: 9.0 cm, VWR, Radnor, PA, USA). After dissolving in tetrahydrofuran and precipitating in cold diethyl ether for three times, the synthesized polymers were dried overnight under vacuum. The same procedure was used for the polymerization of NIPAAm and BA at the 80:20, 82:18, and 83:17 ratios,

aside from the differences in the NIPAAm:BA molar ratios to give a total amount of 20 mmol for both monomers.

To confirm successful synthesis of the pNIPAAm-BA polymers, 10 mg of each polymer was dissolved in deuterated chloroform (CDCl$_3$) and analyzed by $^1$H-NMR with an Ultrashield Plus 600 MHz NMR spectrometer (Bruker, Billerica, MA, USA). The gel permeation chromatography (GPC) experiments were performed with a Waters (Milford, MA, USA) GPC instrument equipped with a 2414 refractive index (RI) detector and Styragel columns (HR4: 5 µm, 7.8 × 300 mm and HR5: 5 µm, 7.8×300 mm) in the presence of tetrahydrofuran as the eluent at a flow rate of 1 ml min$^{-1}$ at 35 °C. Polystyrene standard kits (Mid-High MW, Waters) were used for calibrating and calculating the molecular weight of the polymers. The data were plotted with Origin 8.0 (OriginLab Corporation, MA, USA).

## Measurement of LCST

To determine the LCST of the pNIPAAm-BA polymers, microscopic images of the aqueous samples of the polymers during cooling on a Linkam (Waterfield, UK) temperature-controlled microscope stage were taken. Briefly, a polymer film was prepared by dissolving 10 mg of polymer in 1 ml of ethanol and adding to the Linkam quartz crucible sample holder (Waterfield, UK) for drying. Then 500 µl of deionized water (DW) was added into the crucible of the Linkam temperature-controlled microscope stage mounted on a Zeiss Axio Scope A1 microscope (Oberkochen, Germany). After loading the holder with the sample into the stage, the stage was held at 10 °C for 5 min for temperature equilibration in the sample. Then, the sample was cooled from 10 to −15 °C at a cooling rate of 0.5 °C min$^{-1}$, during which the sample images were taken every 20 s. Grayscale values of the images were quantified using the Image Processing Toolbox in MATLAB 2020b (MathWorks Inc., Natick, MA, United States). Grayscale values of all images were normalized to that of the first image at 10 °C and plotted against the recorded temperature during cooling. The change in grayscale value indicates a change in transparency in the sample. The LCST was determined as the temperature at the point where the second derivative of the data (i.e., the image grayscale value versus temperature) was equal to zero, or the inflection point in the curve of the image grayscale value versus temperature. The data were plotted with Origin 8.0.

## Synthesis of nanoparticles

CPT&siR CRNPs were prepared by a double-emulsion (water-in-oil-in-water or W-O-W) method. Briefly, to prepare the first water phase, 1 mg ml$^{-1}$ aqueous chitosan solution was made by dissolving chitosan in 25 mM sodium acetate (pH 5.0). A total of 200 µl of chitosan solution was added into 100 µl of aqueous sodium chloride (NaCl) solution (20 mg ml$^{-1}$) and mixed. During mixing, 50 µl of siRNA (100 µM) was added dropwise into the mixture and incubated for 1 h at room temperature. For the oil phase, 15 mg of PLGA, 30 mg of pNIPAAm-BA, 10 mg of PF-127, 5 mg of DPPC, and 2.5 mg of CPT were dissolved in 2 ml of dichloromethane (DCM). To form the first W-O emulsion, the first water phase containing the siRNA was added dropwise into the oil phase and emulsified at an amplitude of 10% for 1 min using a Branson 450 sonifier (Hampton, NH, USA). The first emulsion was then added into 5 ml of the second water phase, which was 2% PVA solution containing 10 mg of CS-PF127. The resultant sample was further emulsified by sonication (10% amplitude) for another 1 min to form the W-O-W emulsion, which was stirred overnight at 1200 rpm using a stir bar. Right after sonication, 5 ml of 2% PVA solution was added into the second emulsion. The sample was centrifuged (13800 g) at room temperature for 20 min to collect the CPT&siR CRNPs, which were washed with DW once and redispersed in DW. The procedures for making the CPT CRNPs, siR CRNPs, and CPT&siR PLGA nanoparticles were the same as that for making the

CPT&siR CRNPs except that siR, CPT, and pNIPAAm-BA were not used, respectively. For the preparation of CPT&siR CRNPs (no NaCl), the procedure was the same with that for preparing the CPT&siR CRNPs except that NaCl was not used.

## Characterization of nanoparticles

To determine the encapsulation efficiency (EE) and loading capacity (LC) of CPT, the pellets of the nanoparticles were dissolved in DMSO. The absorbance of CPT was measured by a Tecan (Männedorf, Switzerland) Spark Multimode Microplate Reader at 375 nm and the amount of drug was quantified by a calibration curve made with various known concentrations of CPT. EE and LC were calculated as: EE = (CPT in the pellet)/(CPT fed for synthesis) × 100%, LC = (CPT in the pellet)/(Total weight of the pellet) × 100%. For quantifying the amount of siR in the nanoparticles, siR was isolated by dissolving the pellets with chloroform and extracting the siR with diethylpyrocarbonate (DEPC)-treated tris(hydroxymethyl)aminomethane-ethylenediamine-tetraacetic acid (TE) buffer. The concentration of isolated siR was analyzed by NanoDrop 2000c (Thermo Fisher Scientific).

The phase transition of the CPT&siR PLGA NPs and CPT&siR CRNPs was examined by immersing their aqueous samples in sealed glass vials into cold NaCl solution (−4 °C) for 10 min, followed by gradually warming them up in air to room temperature. Visual appearances of the samples during the cooling-warming treatment were studied by taking photographs. The LCST of nanoparticles in the aqueous sample was also determined by the Linkam temperature-controlled microscope stage in the same way as mentioned above for the pNIPAAm-BA polymers.

The hydrodynamic diameter, zeta potential, and polydispersity index (PDI) of the nanoparticles in DW at a concentration of 0.1 mg ml$^{-1}$ before and after cold treatment were measured by dynamic light scattering (DLS) using a Malvern (Cambridge, UK) Zetasizer Nano ZS instrument. The data of hydrodynamic diameter were plotted with Origin 8.0. The morphology of these nanoparticles with/without cold treatment was examined by a JEOL (Akishima, Tokyo, Japan) JEM 2100 LaB6 TEM after staining with uranyl acetate solution (2%, w w$^{-1}$) and a Hitachi (Chiyoda City, Tokyo, Japan) SU-70 Schottky field emission gun scanning electron microscope after drying in the chemical hood. Moreover, the stability of CPT&siR CRNPs in PBS (0.1 mg ml$^{-1}$) for up to 5 days was examined by measuring their hydrodynamic diameter and PDI.

The stability of siR in CPT&siR CRNPs (1.0 mg ml$^{-1}$) after incubation with PBS for 0, 4, 8, 12, and 24 h was measured by gel retardation assay using 3.0% agarose gel comprising GelRed® Nucleic Acid Gel Stain (Biotium, Fremont, CA, USA). The gel retardation assay was performed in 1× Tris-Borate-EDTA (TBE, Fisher Scientific, Hampton, NH, USA) running buffer at 120 v for 15 min using a Bio-Rad (Hercules, CA) Horizontal Electrophoresis Systems. The unprocessed scans of the gels are provided in the Source Data file.

## Characterization of cold-triggered drug/gene release

To assess the in vitro cold-triggered drug release profile, aliquots of 10 mg of CPT&Cy5-siR CRNPs were dispersed in 3 ml of acetate buffer (pH 5.0), phosphate buffer (pH 6.5), and phosphate buffer (pH 7.4) in a 15-ml centrifuge tube, respectively, and then placed in an Incubating Orbital Shaker (VWR, Radnor, PA, USA) at 100 rpm and 37 °C. For cold treatment at 8 h, the samples were immersed into cold NaCl solution (−4 °C) for 10 min. At various times, 400 µl of the supernatant of the samples were collected for release measurement after centrifuging the samples at room temperature or 0 °C (only for the time point right after the cold treatment) at 13800 g for 20 min. A total of 400 µl of fresh buffer was added into the sample right after each collection of the supernatant. The fluorescence intensity of CPT in the collected supernatant was measured using the Spark Multimode Microplate

Reader, with an excitation wavelength at 370 nm and an emission wavelength of 434 nm. The amount of Cy5-siR released at different time points was also quantified by the Spark Multimode Microplate Reader with an excitation wavelength of 651 nm and an emission wavelength of 670 nm. The cold-triggered release of siR from the nanoparticles at −4, 22, and 37 °C was also investigated with the aforementioned gel retardation assay. The unprocessed scans of the gels are provided in the Source Data file.

## In vitro cellular uptake and intracellular distribution

For the cellular uptake study, GFP$^+$ EO771 cells and normal lymphocytes form mouse spleen were seeded on collagen-coated coverslips in 12-well plate or 12-well plate only at a density of $1 \times 10^5$ cells per well and incubated for 12 h. After removing the culture medium, cells were treated with PBS, free CPT&Cy5-siR, or CPT&Cy5-siR CRNPs at 37 °C for 8 h. The concentrations of CPT and Cy5-siR were 10 μg ml$^{-1}$ and 50 nM, respectively. For confocal microscopy analysis, the treated cells were washed three times with PBS, fixed with 4% paraformaldehyde (PFA) for 15 min at room temperature, and stained with 6-diamidino-2-phenylindole (DAPI, 300 nM) for 10 min at room temperature. Before imaging with a Zeiss LSM 710 confocal microscope, the samples were mounted on glass slides with the mounting medium (CC/Mount™, Sigma-Aldrich). The images were collected with the Zeiss ZEN 2011 SP7 FP3 (black) and analyzed with the Zeiss ZEN 2.6 lite. To quantify cell uptake, the treated cells were detached with trypsin, fixed with 4% PFA, and analyzed by a BD (Franklin Lakes, NJ, USA) FACSCelesta flow cytometer with the built-in software BD FACSDiva™ 8.0.1.1 for data collection.

To visualize the intracellular distribution, EO771 cells grown on collagen-coated coverslips as aforementioned were incubated with PBS, free CPT&Cy5-siR, or CPT&Cy5-siR CRNPs (without or with cold treatment) at the same concentration of Cy5-siR (50 nM) for up to 8 h. At 8 h, cells for the cold treatment group (CPT&Cy5-siR CRNPs+C) were washed with PBS, added with fresh medium, and transported into the chamber of a SP (Gardiner, NY, USA) VirTis AdVantage Pro Freeze Dryer/Lyophilizer, where the temperature was maintained at −4 °C. After 10 min, the samples were taken out of the chamber and warmed back passively to room temperature in a biosafety cabinet. Before mounting to glass slides for imaging by the confocal microscope, the samples were stained with 50 nM LysoTracker™ Green DND-26 (Invitrogen), fixed with 4% PFA, and stained with DAPI. To determine the intracellular distribution of CPT&Cy5-siR CRNPs (no NaCl) in the presence or absence of cold treatment, same procedures were performed except for substituting the CPT&Cy5-siR CRNPs with CPT&Cy5-siR CRNPs (no NaCl).

## In vitro gene silencing

To investigate gene silencing, GFP$^+$ EO771 cells were seeded on collagen-coated coverslips in a 6-well plate at a density of $1 \times 10^5$ cells per well. After overnight incubation, cells were treated with PBS, free GFP-siR, GFP-siR CRNPs, GFP-siR CRNPs+C, and scrambled (Sc)-siR CRNPs+C at the same concentration of GFP-siR (50 nM). At 8 h, cells for the groups with cold treatment were placed into the −4 °C chamber as mentioned above and incubated at −4 °C for 10 min before warming up to room temperature and putting back in the incubator. After further incubation for 40 h, cells were collected and analyzed by flow cytometry or confocal microscopy. For the latter, cells were washed with PBS three times and stained with 50 nM LysoTracker™ Red DND-99 (Invitrogen) and DAPI using the same staining method as described above. Furthermore, the same procedure was used for investigating the downregulation of PD-L1 expression, except substituting the GFP-siR with PD-L1-siRNA (siR) and the cells were stained with PE anti-mouse CD274 at cold (ice-water) temperature for 20 min before flow cytometry analysis.

## In vitro anticancer effect of CPT

Viability of EO771 cells after treatment with blank CRNPs, free CPT&siR, CPT&siR CRNPs, and CPT&siR CRNPs+C at increasing concentrations of CPT ranging from 0 to 40 μg ml$^{-1}$ was examined using the Cell Counting Kit-8 (CCK-8, Dojindo, Rockwill, MD, USA) assay. The siR concentration was kept at 50 nM. Briefly, EO771 cells were seeded in 96-well plates at a density of $2 \times 10^4$ cells per well and cultured for 12 h. The cells were treated with various formulations and incubated for 24 h. For the cold treatment, after incubation with the CPT&siR CRNPs for 8 h, the plate was placed into the −4 °C chamber and maintained at −4 °C for 10 min before warming up to room temperature and putting it back in the incubator. At 24 h, cells were washed with PBS and added with 100 μl of fresh medium. Then, 10 μl of CCK-8 was added into each well and incubated for 1 h. Absorbance was then measured using the Spark Multimode Microplate Reader at 460 nm. To determine cell apoptosis and necrosis, EO771 cells ($2 \times 10^5$ cells per well) were seeded in 6-well plates. After overnight incubation, the cells were treated with PBS, free CPT&siR, CPT&siR CRNPs, or CPT&siR CRNPs+C. The CPT and siR concentrations were 10 μg ml$^{-1}$ and 50 nM, respectively. For the cold treatment, it was done at 8 h as aforementioned. At 24 h, cells were collected and stained with a FITC-Annexin V Apoptosis Detection Kit I (BD) according to the protocol provided by the manufacturer. The samples were analyzed by flow cytometry (BD).

## Measurement of ICD in vitro

To determine the generation of ICD, expression of DAMP molecules including HMGB1, CRT, HSP-70, and HSP-90 was investigated by flow cytometry. Briefly, EO771 cells ($2 \times 10^5$ cells per well) were seeded in 6-well plates and incubated overnight. The medium was changed to fresh medium containing PBS, free CPT&siR, CPT CRNPs, siR CRNPs, or CPT&siR CRNPs. The CPT and siR concentrations were 10 μg ml$^{-1}$ and 50 nM, respectively. After 8 h, cells for the groups with cold treatment were placed into a cold chamber maintained at −4 °C or −20 °C for 10 min. After warming back to room temperature, the cells were put back in the incubator for further culture. At 24 h, cells were collected, fixed with 4% PFA for 10 min, and incubated with primary antibodies of CRT, HMGB1, HSP-70 and HSP-90 overnight. After washing with PBS and further incubated with corresponding fluorescence-tagged secondary antibodies, all the samples were analyzed by flow cytometry.

## In vitro maturation of BMDCs and activation of CD8$^+$ T cells

To investigate the maturation of BMDCs, EO771-OVA cells ($1 \times 10^5$) were seeded on the top wells (i.e., inserts) of transwells and treated with PBS, free CPT&siR, CPT&siR CRNPs and CPT&siR CRNPs+C for 8 h. The CPT and siR concentrations were 10 μg ml$^{-1}$ and 50 nM, respectively, and the cold treatment at −20 °C for 10 min was done at 8 h as aforementioned. Afterward, immature BMDCs were seeded in the bottom wells of the transwells at $1 \times 10^6$ cells per well. At 24 h, the BMDCs were collected and stained with Alexa Fluor® 700 anti-mouse CD3 (for T cell exclusion), PE anti-mouse CD11c, FITC anti-mouse CD86, and APC anti-mouse H-2Kb (bound to SIINFEKL) antibodies for analyzing with flow cytometry.

To determine T cell proliferation, OT-I CD8$^+$ T cells ($1 \times 10^5$) that could recognize the OVA antigen were labeled with CellTrace™ Violet Cell Proliferation Kit (Thermo Fisher Scientific) according to the manufacturer's instruction and co-cultured for 3 days with BMDCs ($1 \times 10^5$) pre-incubated with EO771-OVA cells with the various treatments for 24 h given above in 96-well plate. Proliferation of CD8$^+$ T cells was measured by flow cytometry. For the detection of T cell activation, non-labeled OT-I CD8$^+$ T cells ($1 \times 10^5$) were co-cultured with the aforementioned pre-incubated BMDCs ($1 \times 10^5$) in 96-well plate. After 3 days, cells were stimulated with Cell Stimulation Cocktail (eBioscience, San Diego, CA, USA) for 4 h, stained with Alexa Fluor® 700 anti-mouse CD3, APC/Cyanine7 anti-mouse CD8a, and FITC

anti-human/mouse GZMB antibodies, and analyzed by flow cytometry. The co-culture medium was also collected to examine the secretion of IFN-γ and TNF-α using ELISA kits (Biolegend) by following the protocols provided by the manufacturer.

### In vitro anticancer capability via activating CD8$^+$ T cells

EO771-OVA cells labeled with CellTracker™ Green CMFDA Dye (0.5 μM) were seeded in μ-Dish$^{35mm, low}$ (ibidi, Fitchburg, WI, USA) at a density of $1 \times 10^4$ cells per dish and placed in the cell-incubator stage (mounted on the Zeiss LSM710 microscope) maintained at 37 °C with 5% $CO_2$. The CD8$^+$ T cells ($2 \times 10^3$ per dish) activated as aforementioned were added into the dishes with EO771-OVA cells and monitored for 4 h under microscope. At the end of the study, cells were harvested, stained with propidium iodide (PI), and further analyzed by flow cytometry.

### In vivo biodistribution of CRNPs

The in vivo distribution of CRNPs was investigated in C57BL/6 mice bearing EO771 orthotopic tumor. Briefly, $1 \times 10^6$ of EO771 cells per mouse were administered into the left abdominal mammary fat pad of C57BL/6 mice. When the volume of tumor reached approximately 100 mm$^3$, mice were randomly distributed into three groups and intravenously injected with PBS, free CPT&Cy5-siR, or CPT&Cy5-siR CRNPs. The doses of CPT and Cy5-siR were 2.0 mg kg$^{-1}$ and 30 μg kg$^{-1}$, respectively. At 0, 4, 8, 12, and 24 h, whole-animal images of all mice were acquired using a PerkinElmer (Waltham, MA, USA) IVIS instrument. Moreover, circulation of CRNPs in blood was also studied by collecting blood from the mice at various time points and imaged by the IVIS system. At the end of the study, mice were euthanized and their critical organs including hearts, livers, spleens, lungs, kidneys, and tumors were harvested and imaged by IVIS.

### In vitro hemolysis assay

Whole blood collected from C57BL/6 mice was centrifuged at 500 g and washed with PBS for 3 times. A total of 0.5 ml of whole blood cells (4% in PBS) were mixed with 0.5 ml of DW, PBS, or CPT&siR CRNPs at various CRNP concentrations from 25 to 800 μg ml$^{-1}$. After incubation for 24 h at 37 °C, the samples were centrifuged at 500 g for 5 min. Supernatants were collected and their absorbances at 540 nm were measured by the Spark Multimode Microplate Reader. In addition, after dehydration by a series of ethanol solutions (50%, 75%, 85%, 95%, and 100%), the morphology of erythrocytes after incubation with CPT&siR CRNPs at 800 μg ml$^{-1}$ was compared to that from the PBS group via taking their SEM images using a Hitachi SU-70 Schottky field emission gun scanning electron microscope by following the procedure provided by the manufacturer.

### In vivo antitumor capability through cryo-immunotherapy

To evaluate the antitumor cryo-immunotherapy effect, $1 \times 10^6$ of EO771 cells per gland were administered into both the left and right abdominal mammary fat pad of C57BL/6 mice. On day 10, mice with EO771 tumors on both sides were randomly divided into ten groups (five groups without cryosurgery/cryo and five groups with cryosurgery done on the primary tumors on the left side of the mice) and intravenously administered with PBS (no cryosurgery/Cryo), free CPT&siR (no Cryo), CPT CRNPs (no Cryo), siR CRNPs (no Cryo), and CPT&siR CRNPs (no cryo); and PBS (Cryo), free CPT&siR (Cryo), CPT CRNPs (Cryo), siR CRNPs (Cryo), CPT&siR CRNPs (Cryo). The number (n) of mice for each group is 9 with 6 mice for monitoring the tumor growth and 3 mice for analyzing the immune cells and cytokines on day 20. The doses of CPT and Cy5-siR were 2.0 mg kg$^{-1}$ and 30 μg kg$^{-1}$, respectively. Cryosurgery was done on the left tumors at 8 h after the first injection of the various formulations. The injections were continued every 3 days until the end of the study. Body weight and tumor size were recorded every 3 days, and tumor volume was calculated as follows: $V = 0.5 \times L \times W^2$, where $V$, $L$, and $W$ are the tumor volume, long

diameter, and short diameter, respectively. The maximal tumor size/burden permitted by the IACUC at the University of Maryland, College Park, is 1.5 cm in long/short diameter. During the whole experiment, the maximal tumor size/burden was not exceeded. For cryosurgery, mice were anesthetized with isoflurane first. Cryosurgery was then performed by touching the skin of the tumor area with a cryoprobe (Dahai Mechanical & Electronic Equipment Manufactory, Jiangsu, China) at its central location of the tumor area and freezing for a total of 10 min. The cryoprobe was connected to a DH-286 Cryogenic Therapeutic Apparatus (Dahai Mechanical &Electronic Equipment Manufactory). Two K-type thermocouples were placed at the center and boundary of the tumor area for monitoring the temperature at the two locations, and the freezing was turned on and off to maintain the temperature at tumor center and boundary at approximately −20 and −4 °C, respectively.

On day 20, 3 mice per groups were euthanized for harvesting tumors and blood for further analysis. Briefly, the obtained tumors were minced into small pieces and digested with DNAse I (0.1 mg ml$^{-1}$, Roche, Basel, Switzerland) and collagenase IV (2.0 mg ml$^{-1}$, Roche) for 1 h at 37 °C. Before analyzing by flow cytometry, the single cell suspensions were filtered through a 70-μm strainer and stained and gated with a combination of markers: dendritic cells (CD3$^-$CD11c$^+$CD86), Treg cells (CD3$^+$CD4$^+$Foxp3$^+$), activated CD8 (CD3$^+$CD8$^+$GZMB$^+$), tumor-associated macrophage (CD45$^+$CD11b$^+$Ly6C$^-$Ly6G$^-$F4/80$^+$) M1 (F4/80$^+$CD206$^-$CD86$^+$), tumor-associated macrophage M2 (F4/80$^+$CD206$^+$CD86$^-$), monocytic myeloid-derived suppressor cells (M-MDSC, CD11b$^+$Ly6C$^+$Ly6G$^-$), and polymorphonuclear myeloid-derived suppressor cells (PMN-MDSC, CD11b$^+$ Ly6C$^-$Ly6G$^+$). The examination of immune cells in blood was performed by removing the erythrocytes with RBC lysis buffer (Biolegend) and staining with antibodies similarly as mentioned above. For analysis of memory immune response, the collected blood and spleen cells were stained and gated with a combination of markers for CD8$^+$ effector memory T (T$_{EM}$, CD3$^+$CD4$^-$CD8$^+$CD62L$^-$CD44$^+$) cells and central memory T (T$_{CM}$, CD8$^+$CD44$^+$CD62L$^+$) cells. All flow cytometry data was analyzed with FlowJo (v10, BD). Moreover, after cryosection, the presence of CD8$^+$GZMB$^+$ cells and the generation of HMGB1, CRT, HSP-70, and HSP-90 in tumor tissues were investigated by immunofluorescence staining. Briefly, cryosectioned tumor slides were washed with PBS for 3 times, fixed with 4% PFA for 10 min, and rinsed with PBS for 3 times. Afterward, the samples were permeabilized with 0.2% Triton X-100 for 5 min, washed with PBS for 3 times, and incubated with 5% BSA for 1 h at room temperature. After removing the BSA with 3 times of PBS washing, the samples were incubated with the primary antibodies overnight at 4 °C. After 3 times of PBS washing, the samples were incubated with the corresponding fluorescence-labeled secondary antibodies for 30 min at room temperature. The samples were further stained with DAPI, washed with PBS, and mounted on coverslips for imaging with the Zeiss LSM 710 microscope. Furthermore, the expression of PD-L1 in in vivo tumors after various treatments was determined by western blotting. Briefly, tumor tissues were weighed, minced, and lysed with 1× radioimmunoprecipitation assay (RIPA) buffer (Cell Signaling Technology, Danvers, MA, USA) containing 1 mM phenylmethylsulfonyl fluoride (PMSF) (Cell Signaling Technology). Supernatants of samples were collected after centrifuging at 13,200 g and 4 °C for 30 min. The total proteins were separated with 10% SDS-PAGE gel and then transferred onto the polyvinylidene difluoride (PVDF) membrane (0.45 μm, MilliporeSigma, Burlington, MA, USA). After blocking with 5% bovine serum albumin (BSA) and washing using phosphate-buffered saline with Tween 20 (PBST), the membranes were incubated with the primary antibodies of PD-L1 overnight. Afterwards, the membranes were incubated with corresponding HRP-conjugated secondary antibodies for 1 h, immersed in Pierce™ ECL Western Blotting Substrate (Thermo Fisher Scientific), and scanned by the FluorChem E System Gel Imaging System

(ProteinSimple, San Jose, CA, USA). The unprocessed scans of the blots are provided in the Source Data file.

At the end of the study, mice were euthanized and critical organs including hearts, livers, spleens, lungs, kidneys, and tumors were harvested from the mice. Tumors were weighed. H&E staining was conducted for paraffin-sectioned slices of all the organs by following the standard procedure. Tumors were also cryosectioned by following the standard procedure for immunostaining of CD31 and Ki-67. The amount of alanine aminotransferase (ALT) and aspartate aminotransferase (AST) levels in the blood serum of mice collected at the end of the study were also measured by Mouse ALT ELISA Kit (Abcam) and Mouse AST ELISA Kit (Abcam) using the protocols provided by the manufacturer.

### In vivo anti-metastatic effect of ICIE

To examine the long-term antitumor memory immune response of ICIE, Balb/c mice were inoculated with 4T1 cells into the abdominal mammary fat pad of mice at $1 \times 10^6$ cells per gland. After 5 days, mice with orthotopic 4T1 tumors were randomly grouped and injected with various formulations including: PBS, PBS+cryo, CPT&siR CRNPs, and CPT&siR CRNPs+cryo. The number of mice in each group is 8, with another group of 8 mice named PBS without orthotopic tumor as the control for metastatic induction only. The doses of CPT and Cy5-siR were 2.0 mg kg$^{-1}$ and 30 μg kg$^{-1}$, respectively. Cryosurgery was done on the orthotopic tumors at 8 h after the first injection of the various formulations. All injections were continued every 3 days until the death of the animals or the end of the study. To establish the in vivo lung metastatic model, each Balb/c mouse was intravenously injected with $1 \times 10^5$ 4T1 cells 10 days post the first injection of different formulations. After monitoring for 15 days, all live mice were euthanized for the collection of orthotopic tumors and lungs. The orthotopic tumors and lungs were also collected from animals when they were found dead before the 15 days of monitoring. H&E staining was conducted for paraffin-sectioned slices of lungs by following the standard procedure.

### Statistical analysis

All quantitative data are presented as mean ± standard deviation (S.D.) from three or more independent runs. One-way analysis of variance (ANOVA) with Tukey's post-test and correction for multiple comparisons was used when multiple groups with one independent variable/factor (e.g., treatment) were compared. Two-way ANOVA with Sidak's post-test and correction for multiple comparisons was used when multiple groups with two independent variables/factors (e.g., treatment and dose). The survival benefit was analyzed by Log-rank (Mantel-Cox) test. Statistical analysis and plotting were carried out by using the GraphPad (San Diego, CA, USA) Prism 9 software, unless it is specifically mentioned otherwise above (i.e., plotted with Origin 8.0). A $p$ value less than 0.05 indicates a statistically significant difference.

### Reporting summary

Further information on research design is available in the Nature Portfolio Reporting Summary linked to this article.

## Data availability

Source data are provided with this paper. The data generated in this study are available within the Article, Supplementary Information, or Source Data file. Source data are provided with this paper.

## Code availability

The code used for analyzing the LCST is provided with this paper as Supplementary Code file.

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

## Acknowledgements

This work was partially supported by a grant from the National Institutes of Health (NIH R01CA243023 to XH). We would like to thank Bin Jiang for his help with the H&E staining experiment.

## Author contributions

X.H. and W.O. conceived the project; X.H. supervised the study; X.H. and W.O. designed experiments; W.O. conducted experiments with assistance from S.S., A.M.W., E.A.K., J.X., Y.F., and J.G.S.; X.H., W.O., and S.S. analyzed the data; W.O. wrote the manuscript draft; X.H., S.S., A.M.W.,

J.G.S, C.X., S.N., N.P.T., X.L., and K.H.R.T. edited the manuscript; and all authors approved the manuscript.

## Competing interests

The authors declare the following financial interests/personal relationships which may be considered as potential competing interests: The authors (X.H. and W.O.) disclosed the technology reported in this work to the University of Maryland Office of Technology Commercialization. The remaining authors declare no other competing interests.
