## [Peer Review File · Nature Communications]

In-situ cryo-immune engineering of tumor microenvironment
with cold-responsive nanotechnology for cancer
immunotherapyREVIEWER COMMENTS

Reviewer #1 (Remarks to the Author): with expertise in nanoparticles, RNA delivery, cancer immunology

In this manuscript, He and co-authors report a novel cancer treatment strategy that combined cryosurgery with cold-responsive nanoparticles (CRNPs). CRNPs contains anti-cancer drug irinotecan and anti-PD-L1 siRNA to promote cancer immunotherapy. The authors studied DC maturation and their antigen-presenting ability, as well as T cell activation after dual-treatment. Further, the authors found that CRNPs accumulates in the tumor after intravenous injection and rapidly releases irinotecan and siRNA upon cryosurgery. Lastly, the dual-treatment strategy stimulates antitumor immune response and slowed tumor growth in both treated and distal tumor. The research design is very innovative, and the findings are comprehensive. Below are minor comments.

1. Page 3, line 62, it is unclear why the authors mentioned ultrasonography and what is the reason for incomplete tumor destruction. Please clarify the information.
2. Figure 2e and Supplemental Figure 5, the authors concluded that NaCl accounts for the endo/lysosomal escape of the CRNPs. Could the authors discuss the underlying mechanism for this finding, and add it to the manuscript?
3. Please add value on the scale bar in Figure 4A, 4C and 4D.
4. Figure 5b, in the illustration of the in vivo experiments, was the cryosurgery only performed once at 8h after the first i.v. injection? How long can the low temperature remain in the tumor area?

Reviewer #2 (Remarks to the Author): with expertise in thermo-responsive nanoparticles, immunotherapy

This interesting and clear manuscript by Ou et al. addresses the use of in situ cryo-immune engineering strategy to limit tumor growth and enhance anti-tumor immunity. The manuscript is of interest, nicely introduced, cryosurgery become increasingly popular for cancer therapy. The authors have used cold-responsive nanoparticles with the ability to rapidly release both anticancer drug and anti-PD-L1 siRNA specifically into the cytosol upon cryosurgery and lead to potent immunogenic cell death to reverse the immunosuppressive TME, thus promoting the host's immune system to recognize and attack tumors. The authors have used several very well-defined experimental systems, demonstrating that situ cryo-immune engineering strategy has an impact on only primary but also distant tumors, and might complement other cancer immunotherapy-based strategies. The manuscript includes relevant and supplementary information that complements the main text is solid. However, a number of issues could be addressed for the manuscript to gain robustness and to fully support the author's conclusions.

General comments:

As a co-delivery system, CPT and siR from CPT&siR CRNPs show similar cellular release behavior upon cryosurgery, the authors should add the data about the optimal formulation of CPR-to-siR ratios in CPT&siR CRNPs. Besides, does CPT affect the effect of siR or siR affect the effect of CPT? The authors should assess the synergistic effect of co-delivery of CPR and siR. As an example, authors should add CPT in the treated group (In fig2 f-I).

Cryosurgery is done by cooling to cause ice formation (i.e., frostbite) in tumors, it is so important to target tumor cells. The authors confirm the successful uptake of CPT&siR CRNPs in breast cancer EO771 cells, but how about the uptake of CPT&siR CRNPs by other cells? Authors should add the evidence of target ability of CRNPs between tumor cells and other cells.

In fact, cryosurgery can not only act on tumor cells. A major issue that the impact of cryosurgery on other immune cells like DCs, M2, T cells, and MDSCs. Is possible that immune cells in TME also frostbite upon cryosurgery?

Cytokines are major regulators of innate and adaptive immunity that enable cells of the immune system to communicate over short distances and boost immune responses. In situ cryo-immune engineering (ICIE) strategy is a feasible alternative strategy for boosting the antitumor immune responses, but cytokines production from immune cells like DCs and TME is not clear in this manuscript. For instance, there is no clear data on the production of pro-inflammatory cytokines such as tumor necrosis factor (TNF)- α and interferon (IFN)- γ , in the experiment of BMDCs after co-culturing them with EO771-OVA cells.

In EO771-OVA tumor model, ICIE strategy successfully reverses immunologically “cold” TME into a “hot” one. As a whole, can the authors consider ICIE applied to other tumor models? CPT and anti-PD-L1 siRNA are the common drugs. If applied ICIE in other models, it will be more fully support the potential of ICIE in the future. Moreover, generating durable immune memory to a specific tumor antigen is necessary for cancer immunotherapy, can the authors check the prevention of metastasis and the long-term immune memory effect of the possibility of ICIE in EO771-OVA, EO771, or others?

Reviewer #3 (Remarks to the Author): with expertise in cryotherapy, breast cancer, immunology

Recommendation: Major Revision

The authors developed a novel cryo-immunotherapeutic strategy by using cryosurgery and cold-responsive nanomaterials loaded with CPT and PD-L1 siRNA. They claimed that ICIE potentiates ICD, reverses the immunosuppressive TME, and downregulates PD-L1 expression in breast cancer cells, thus generating a robust antitumor and long-term memory immune response for eradicating both primary and distant tumors in vivo. The work is original, comprehensive and meaningful. But there are still some concerns of model and methodology used. Further, the underlying mechanism of observed phenomena also need to be addressed.

Specific Comments:

1. Mouse EO771 breast tumor model is considered as a highly immunogenic model and is poorly metastatic when compared to other models. It might be better addressed that ICIE can reverse local and systematic immunosuppression by repeating the phenotype in the much more immunosuppressive model, i.e. 4T1 in Balb/c background.
2. Can ICIE prolong the survival time of mice? The survival curve of mice bearing one tumor (primary only) and two tumors (primary and distant) treated by various therapies should be plotted. In addition, if any individuals survived after treatment, EO771 tumor cell rechallenge could be used to evaluate the antitumor immune memory.
3. Fluorescence imaging showed the tumor accumulating/targeting capability of the CRNPs in Fig. 4d-e, and the authors did confirm that ICIE induces no evident damage in major organs in Supplementary Fig. 21. However, the accumulation of CRNPs in liver is significantly more than that in tumor as shown in Fig. 4d-e, which may have potential toxicity in consideration of clinical translation.
4. Cryosurgery was done only once (blue arrow, Fig. 5b) at 8 h after the first injection of the formulations. What is the purpose and function of the following injection of formulations after cryosurgery, since these particles are cold responsive?
5. Fig.3a, the authors concluded that the formulation of CPT&siR CRNPs with cold treatments (i.e., ICIE) at -4 and -20 °C induced higher expression of HSP-70, HSP-90, HMGB1, and CRT than all the other formulations with the same cold treatment (line 196-199). But it seems that only the increase of HMGB1 at -4 and HSP-70 at -20 °C are significant. The authors should calculate the grayscale value of western blot protein bands and perform a statistical analysis.
6. In Supplementary Fig.13e-f and Supplementary Fig.16b there are essentially no Tregs to be seen in most groups. The Foxp3 stain is close to background, while the Foxp3 staining in Fig. 6a, Supplementary Fig.15g, Supplementary Fig.16a is good. Overall, why are the gating strategies of the same cells and molecules different in Supplementary Fig.12-16? It seems that the staining protocol is not standardized.

Reviewer #4 (Remarks to the Author): with expertise in thermo-responsive nanoparticles, immunotherapy

This manuscript is reporting in situ cryo-immune engineering with cold-responsive nanotechnology for cancer immunotherapy. For the cryo immune modulation, cold-responsive nanoparticles were designed with a temperature sensitive pNIPAM-BA polymers by adjusting cold responsive LCST. Combinational therapeutic components including irinotecan and PD-L1 silencing siRNA were integrated into the chitosan surface functionalized cold-responsive nanoparticles. During in vitro studies, cold triggered payloads release and endo/lysosomal escape of siR were proved. DAMPs expression and DC maturation following ICIE treatment using the nanoparticles were also demonstrated. Subsequently, CD8+ T cells were activated for potential in vivo tumor cell killing. Finally, the therapeutic response and immune response of the ICIE were presented with breast tumor orthotopic mice model having primary and secondary tumors. The results showed that ICIE treatment with the cold-responsive nanoparticles efficiently modulated the immune suppressive TME and generated anti-cancer immune memory effect, resulting in killing primary tumors and distant tumors at the same time.

The idea of cryo-immune modulation with cold responsive nanoparticles has been proposed with previous reports. The benefits of cryo treatment in immune modulation have proved in recent studies. Thus, various combinational immunotherapy approaches also have been reported. Specific advantages of using cold-responsive nanoparticles are not fully convincing.

(Introduction)

It is not clear if the ultrasonography limitation is the main reason for the incomplete tumor destruction and cancer recurrence.

(Design of nanocarriers)

Quite complex design with multiple components would discouraging to translate the nanotechnology further.

pNIPAM might not be a best choice for in vivo study with their reported potential toxicity.

It is not clear that the synergistic effect of Cryo + irinotecan + PD-L1 silencing siRNA combination therapeutics. It might be difficult to discriminate the synergistic combination effects in each addition of therapeutics.

It is unclear that the use of Chitosan-PF127 and NaCl in the system.

(Therapy)

The role of irinotecan is unclear.

Are those black regions necrotic areas or traces of probe insertion in Fig 7c?

Point-by-point response to reviewers' comments

We would like to thank all the four reviewers for their thoughtful and insightful comments! We have revised our manuscript accordingly. A list of our point-by-point responses to all the comments is given below, and all changes made to address the comments are highlighted in the manuscript text.

Reviewer #1 (Remarks to the Author): with expertise in nanoparticles, RNA delivery, cancer immunology

In this manuscript, He and co-authors report a novel cancer treatment strategy that combined cryosurgery with cold-responsive nanoparticles (CRNPs). CRNPs contains anti-cancer drug irinotecan and anti-PD-L1 siRNA to promote cancer immunotherapy. The authors studied DC maturation and their antigen-presenting ability, as well as T cell activation after dual-treatment. Further, the authors found that CRNPs accumulates in the tumor after intravenous injection and rapidly releases irinotecan and siRNA upon cryosurgery. Lastly, the dual-treatment strategy stimulates antitumor immune response and slowed tumor growth in both treated and distal tumor. The research design is very innovative, and the findings are comprehensive. Below are minor comments.

Re: We appreciate the reviewer for the thoughtful and insightful comments! All the minor comments are carefully addressed in this resubmission as detailed below.

1. Page 3, line 62, it is unclear why the authors mentioned ultrasonography and what is the reason for incomplete tumor destruction. Please clarify the information.

Re: Per the reviewer's advice, we have clarified the reason for incomplete tumor destruction when using medical ultrasonography in the last paragraph on page 3. Medical ultrasonography is based on the hyperechoic property of ice, but the amount of ice formation above ~ -4 °C is too small to detect by ultrasonography. Moreover, ultrasonography can't tell where the temperature is -20 °C that is needed to ensure cancer cell death.

2. Figure 2e and Supplemental Figure 5, the authors concluded that NaCl accounts for the endo/lysosomal escape of the CRNPs. Could the authors discuss the underlying mechanism for this finding, and add it to the manuscript?

Re: Per the reviewer's advice, we have discussed the possible mechanism for the NaCl-mediated endo/lysosomal escape in the last 4 lines of the 3rd paragraph on page 6 and the first 2 lines on page 7 of the revised manuscript. After the CRNPs are internalized into cancer cells in their endo/lysosomes, the cold treatment triggers rapid release of the encapsulated Na^+ and Cl^- from the CRNPs into the endo/lysosomes, resulting in a surge of osmolality (i.e., hypertonicity) in the endo/lysosomes. This causes influx of water, leading to rapid destabilization or rupture of the endo/lysosomes (Ref. 54). The encapsulated CPT and PD-L1 siRNA are then rapidly released into the cytosol to perform the chemotherapy and gene silencing functions.

3. Please add value on the scale bar in Figure 4A, 4C and 4D.

Re: Per the advice, we have added the value on the scale bars in Figure 4a, 4c, and 4d.

4. Figure 5b, in the illustration of the in vivo experiments, was the cryosurgery only performed once at 8h after the first i.v. injection? How long can the low temperature remain in the tumor area?

Re: We appreciate the reviewer's insightful comment! As illustrated in Figure 5b, cryosurgery was performed only once at 8 h after the first i.v. injection in this study. According to the experimental data shown in Figure 5d-e, the low temperature was kept for 10 min in the tumor area. Instead of doing cryosurgery after all injections, we chose to perform the cryosurgery only once so that our in vivo experiment would be similar to the cryosurgery clinically applied to patients. In the clinic, cryosurgery is performed usually once, and it is difficult to perform cryosurgery on patients for multiple times. When combined with our cold responsive nanoparticles, the cold temperature during cryosurgery can effectively induce rapid drug release than non-cold treatment group (Supplementary Fig. 3b-d), resulting in effective damage to tumor cells and silencing of PD-L1 expression to provoke active antitumor immune responses. To ensure long-term tumor eradication and inhibit the immune checkpoint expressed in tumor cells, we continue to inject the nanoparticles that could sustainably release drugs, which is similar to how chemotherapy is done in the clinic. Chemotherapy is usually done via multiple injections in the clinic. This is now clarified in the 3rd paragraph on page 10. We want to thank the reviewer again for all the insightful comments!

Reviewer #2 (Remarks to the Author): with expertise in thermo-responsive nanoparticles, immunotherapy

This interesting and clear manuscript by Ou et al. addresses the use of in situ cryo-immune engineering strategy to limit tumor growth and enhance anti-tumor immunity. The manuscript is of interest, nicely introduced, cryosurgery become increasingly popular for cancer therapy. The authors have used cold-responsive nanoparticles with the ability to rapidly release both anticancer drug and anti-PD-L1 siRNA specifically into the cytosol upon cryosurgery and lead to potent immunogenic cell death to reverse the immunosuppressive TME, thus promoting the host's immune system to recognize and attack tumors. The authors have used several very well-defined experimental systems, demonstrating that situ cryo-immune engineering strategy has an impact on only primary but also distant tumors, and might complement other cancer immunotherapy-based strategies. The manuscript includes relevant and supplementary information that complements the main text is solid. However, a number of issues could be addressed for the manuscript to gain robustness and to fully support the author's conclusions.

Re: We thank the reviewer for the thoughtful and insightful comments!

General comments:

As a co-delivery system, CPT and siR from CPT&siR CRNPs show similar cellular release behavior upon cryosurgery, the authors should add the data about the optimal formulation of CPT-to-siR ratios in CPT&siR CRNPs. Besides, does CPT affect the effect of siR or siR affect the effect of CPT? The authors should assess the synergistic effect of co-delivery of CPT and siR. As an example, authors should add CPT in the treated group (In fig2 f-l).

Re: Per the reviewer's advice, we have added the data on the optimization of CPT to siR in CPT&siR CRNPs. To optimize the formulation, we kept the feeding amount of siR being the same, and varied the feeding amount of CPT (1%, 3% and 5% of the polymers). As shown in the new Supplementary Fig. 2a-b, the EE of CPT decreases monotonically with the increase of the CPT feeding percentage, while the LC of CPT reaches a plateau at the 3% feeding percentage. Therefore, the 3% CPT feeding percentage was used in this work, for which the LC of CPT is $1.5 \pm 0.2\%$ with an encapsulation efficiency (EE) of $51.2 \pm 8.2\%$. This info is now added in the 2nd paragraph on page 5 of this revision.

To assess the synergistic effects of co-delivery of CPT and siR in the CPT&siR CRNPs, we conducted more experiments to compare the tumor attacking capability of CPT&siR CRNPs (either with or without cold treatment) with that of CPT CRNPs alone, siR CRNPs alone, and the simple addition of the effect (i.e., additive effect) of the CPT CRNPs alone and siR CRNPs alone. The data are shown in the new Supplementary Fig. 10. In the absence of cold treatment, co-delivery of CPT and siR in the CPT&siR CRNPs has no synergistic effect and leads to 30.7% of cancer cell death, which is less than the sum value (39.0%, i.e., for the simple additive effect) of the antitumor effect of CPT CRNPs alone and siR CRNPs alone (Supplementary Fig. 10a-b). Importantly, when cold treatment is applied, the synergistic effect of CPT and siR co-delivered with CPT&siR CRNPs is evident (Supplementary Fig. 10c-d), showing significantly higher percentage of cancer cell death (80.5%) than the sum value (61.4%, i.e., for the simple additive effect) of CPT CRNPs alone and siR CRNPs alone (labeled in dotted box). The info is now added in the first paragraph on page 9.

Cryosurgery is done by cooling to cause ice formation (i.e., frostbite) in tumors, it is so important to target tumor cells. The authors confirm the successful uptake of CPT&siR CRNPs in breast cancer EO771 cells, but how about the uptake of CPT&siR CRNPs by other cells? Authors should add the evidence of target ability of CRNPs between tumor cells and other cells.

Re: Per the reviewer's advice, we conducted more experiments to study the uptake of CPT&siR CRNPs in lymphocytes from mouse spleen and the new data are shown in Supplementary Fig. 4d. The results show that the uptake of CPT&Cy5-siR CRNPs in lymphocytes are negligible (~0.9%), while the percentage is 92.8% in EO771 cancer cells. The aforementioned info is now added in the 3rd paragraph on page 6 of this revised manuscript.

In fact, cryosurgery can not only act on tumor cells. A major issue that the impact of cryosurgery on other immune cells like DCs, M2, T cells, and MDSCs. Is possible that immune cells in TME also frostbite upon cryosurgery?

Re: We thank the reviewer for this insightful point! Yes, cryosurgery may act not only on tumor cells but also other immune cells including DCs, M2, T cells and MDSCs in tumor. As shown in Figure 5 & 6, before cryosurgery, the TME is immunologically "cold", with the presence of a high percentage of M2 macrophages, MDSCs, as well as Treg cells that promote tumor growth. Frostbite of these

immunosuppressive cells may be an important mechanism to turn the immunologically “cold” TME into a “hot” one. This is now discussed at the end of the first paragraph on page 11.

Cytokines are major regulators of innate and adaptive immunity that enable cells of the immune system to communicate over short distances and boost immune responses. In situ cryo-immune engineering (ICIE) strategy is a feasible alternative strategy for boosting the antitumor immune responses, but cytokines production from immune cells like DCs and TME is not clear in this manuscript. For instance, there is no clear data on the production of pro-inflammatory cytokines such as tumor necrosis factor (TNF)- α and interferon (IFN)- γ , in the experiment of BMDCs after co-culturing them with EO771-OVA cells.

Re: Per the reviewer’s advice, to investigate the cytokine production from immune cells like DCs and TME, we conducted new experiments to measure the production of TNF- α and IFN- γ after co-culturing BMDCs with EO771-OVA cells. As shown in the new Supplementary Fig. 9, the treatment of CPT&siR CRNPs with cold treatment significantly promotes the secretion of IFN- γ and TNF- α , showing 2.3-, and 1.6-fold higher secretions when compared to the groups without cold treatment, respectively. These findings further confirm that our ICIE strategy is a feasible strategy for boosting the antitumor immune response. This is now added in the 2nd paragraph on page 8.

In EO771-OVA tumor model, ICIE strategy successfully reverses immunologically “cold” TME into a “hot” one. As a whole, can the authors consider ICIE applied to other tumor models? CPT and anti-PD-L1 siRNA are the common drugs. If applied ICIE in other models, it will be more fully support the potential of ICIE in the future. Moreover, generating durable immune memory to a specific tumor antigen is necessary for cancer immunotherapy, can the authors check the prevention of metastasis and the long-term immune memory effect of the possibility of ICIE in EO771-OVA, EO771, or others?

Re: Per the reviewer’s advice, we conducted new experiments to apply ICIE to the 4T1 metastatic tumor model in Balb/c mice and check the prevention of metastasis and the long-term memory immune response. Balb/c mice with 4T1 orthotopic breast tumors were applied with different treatments (including ICIE). After priming for 10 days, a lung metastatic tumor model was formed by intravenous injection of 4T1 cancer cells. The results show that ICIE can not only significantly inhibit the growth of orthotopic 4T1 tumor, but also dramatically increase the percentage of effective memory T cells compared to the CPT&siR CRNPs without cryosurgery and PBS group with orthotopic tumor and cryosurgery by 2.2 and 2.5 folds, respectively (see the new Figure 8). Furthermore, ICIE greatly reduces the formation of metastatic foci in the lung, decreases the tumor burden, and shows less metastasis and more alveolar areas in lung when compared to CPT&siR CRNPs without cryosurgery or PBS groups with/without cryosurgery. These results support that ICIE can be applied to treat other tumor models and generate durable long-term memory immune responses to prevent tumor metastasis. These new results are described in detail in the 1st paragraph on page 15. We want to thank the reviewer again for all the insightful comments!

Reviewer #3 (Remarks to the Author): with expertise in cryotherapy, breast cancer, immunology
Recommendation: Major Revision

The authors developed a novel cryo-immunotherapeutic strategy by using cryosurgery and cold-responsive nanomaterials loaded with CPT and PD-L1 siRNA. They claimed that ICIE potentiates ICD, reverses the immunosuppressive TME, and downregulates PD-L1 expression in breast cancer cells, thus generating a robust antitumor and long-term memory immune response for eradicating both primary and distant tumors in vivo. The work is original, comprehensive and meaningful. But there are still some concerns of model and methodology used. Further, the underlying mechanism of observed phenomena also need to be addressed.

Re: We thank the reviewer for the thoughtful and insightful comments! In this revision, all the concerns and the underlying mechanisms of the observed phenomena are carefully addressed, as detailed below.

Specific Comments:

1. Mouse EO771 breast tumor model is considered as a highly immunogenic model and is poorly metastatic when compared to other models. It might be better addressed that ICIE can reverse local and systematic immunosuppression by repeating the phenotype in the much more immunosuppressive model, i.e. 4T1 in Balb/c background.

Re: Per the reviewer’s advice, we conducted more experiments to apply ICIE to the 4T1 metastatic tumor model in Balb/c mice. Balb/c mice with 4T1 orthotopic tumor were applied with different

treatments (including ICIE). After priming for 10 days, a lung metastatic tumor model was formed by intravenous injection of 4T1 cancer cells. The results show that ICIE can not only significantly inhibit the growth of orthotopic 4T1 tumor, but also significantly increase the percentage of effective memory T cells compared to the CPT&siR CRNPs without cryosurgery and PBS group with primary tumor and cryosurgery by 2.2 and 2.5 folds, respectively (see the new Figure 8). Furthermore, ICIE greatly reduces the formation of metastatic foci in the lung, decreases the tumor burden, and shows less metastasis and more alveolar areas in lung when compared to CPT&siR CRNPs without cryosurgery or PBS groups with/without cryosurgery. These results support that ICIE can be applied to treat other tumor models and generate durable long-term memory immune responses to prevent tumor metastasis. These new results are described in detail in the first paragraph on page 15 of this revision.

2. Can ICIE prolong the survival time of mice? The survival curve of mice bearing one tumor (primary only) and two tumors (primary and distant) treated by various therapies should be plotted. In addition, if any individuals survived after treatment, E0771 tumor cell rechallenge could be used to evaluate the antitumor immune memory.

Re: During the 37-day period of antitumor study, no dead mice were observed for the E0771 primary and distant tumor model. For the experiment with 4T1 metastatic tumor model in Balb/c mice, ICIE significantly prolongs the total survival rate to 73%, while the survival rate in CPT&siR CRNPs without cryosurgery on primary tumor and PBS group with cryosurgery on primary tumor is 50% and 47%, respectively (see the new Supplementary Fig. 27). Further, in Balb/c mice bearing 4T1 primary and metastatic tumors, the effective memory T cells from blood of survived mice were collected and analyzed by flow cytometry. The results show that ICIE can significantly increase the percentage of effective memory T cells compared to the CPT&siR CRNPs without cryosurgery and PBS group with orthotopic tumor and cryosurgery by 2.2 and 2.5 folds, respectively (Fig. 8f). The aforementioned info is now added in the first paragraph on page 15.

3. Fluorescence imaging showed the tumor accumulating/targeting capability of the CRNPs in Fig. 4d-e, and the authors did confirm that ICIE induces no evident damage in major organs in Supplementary Fig. 21. However, the accumulation of CRNPs in liver is significantly more than that in tumor as shown in Fig. 4d-e, which may have potential toxicity in consideration of clinical translation.

Re: In Figure 4d-e, high accumulation of CRNPs in liver can be observed, which is actually not unusual for nanoparticles after administration into the body (Ref. 57). However, in this study, the CRNPs are a kind of cold-responsive nanoparticle. The release of encapsulated drugs is slow in the absence of cold treatment. Since cryosurgery is only performed on tumors, those CRNPs accumulated in liver will be excreted from the body with minimal release of the encapsulated drugs, which should minimize their toxicity to the liver. This is first confirmed by the fact that no evident damage is observable in major organs including liver after the 37-day period of antitumor study (Supplementary Figs. 23-24). To further verify the safety of our CRNPs, we conducted more experiments to measure the alanine aminotransferase (ALT) and aspartate aminotransferase (AST) levels in blood of mice at the end of the study. As shown in the new Supplementary Fig. 25, there is no significant change in the ALT and AST levels in the blood of mice for the treatment of CPT CRNPs, siR CRNPs, or CPT&siR CRNPs. These data again support that CRNPs don't induce any evident toxicity to the liver in vivo. The aforementioned info is now added in the 2nd paragraph on page 9 and the 3rd paragraph on page 14.

4. Cryosurgery was done only once (blue arrow, Fig. 5b) at 8 h after the first injection of the formulations. What is the purpose and function of the following injection of formulations after cryosurgery, since these particles are cold responsive?

Re: Like many other polymeric nanoparticles for sustained drug release, the two agents are also released slowly from our CRNPs (Supplementary Fig. 3b-d) and may have therapeutic benefit for cancer treatment. Unlike cryosurgery that is difficult to do multiple times and usually done once, i.v. injection of the nanoparticles is not difficult to do and multiple injections are commonly done for chemotherapy in the clinic. Therefore, we inject the nanoparticles for multiple times. This is now clarified in the 3rd paragraph on page 10.

5. Fig.3a, the authors concluded that the formulation of CPT&siR CRNPs with cold treatments (i.e., ICIE) at -4 and -20 °C induced higher expression of HSP-70, HSP-90, HMGB1, and CRT than all the other formulations with the same cold treatment (line 196-199). But it seems that only the increase of HMGB1 at -4 and HSP-70 at -20 °C are significant. The authors should calculate the grayscale value

of western blot protein bands and perform a statistical analysis.

Re: To address the comments, we conducted more experiments to quantify the expression of HSP-70, HSP-90, HMGB1, and CRT by the more quantitative flow cytometry method. Fig. 3a is now updated with the new quantitative data and we added the statistical analysis in the figures per the reviewer's advice, which shows the formulation of CPT&siR CRNPs with cold treatments (i.e., ICIE) at -4 and -20 °C induced higher expression of HSP-70, HSP-90, HMGB1, and CRT than all the other formulations with the same cold treatment.

6. In Supplementary Fig.13e-f and Supplementary Fig.16b there are essentially no Tregs to be seen in most groups. The Foxp3 stain is close to background, while the Foxp3 staining in Fig. 6a, Supplementary Fig.15g, Supplementary Fig.16a is good. Overall, why are the gating strategies of the same cells and molecules different in Supplementary Fig.12-16? It seems that the staining protocol is not standardized.

Re: We thank the reviewer for catching the error! We double-checked the figures (now Supplementary Figs. 15-18 in this revision) to ensure the flow data in all figures are analyzed using the same gating strategy. We have also confirmed that all the staining experiments were strictly performed following the standardized protocols provided by the manufacturers. We want to thank the reviewer again for all the insightful comments!

Reviewer #4 (Remarks to the Author): with expertise in thermo-responsive nanoparticles, immunotherapy

This manuscript is reporting in situ cryo-immune engineering with cold-responsive nanotechnology for cancer immunotherapy. For the cryo immune modulation, cold-responsive nanoparticles were designed with a temperature sensitive pNIPAM-BA polymers by adjusting cold responsive LCST. Combinational therapeutic components including irinotecan and PD-L1 silencing siRNA were integrated into the chitosan surface functionalized cold-responsive nanoparticles. During in vitro studies, cold triggered payloads release and endo/lysosomal escape of siR were proved. DAMPs expression and DC maturation following ICIE treatment using the nanoparticles were also demonstrated. Subsequently, CD8+ T cells were activated for potential in vivo tumor cell killing. Finally, the therapeutic response and immune response of the ICIE were presented with breast tumor orthotopic mice model having primary and secondary tumors. The results showed that ICIE treatment with the cold-responsive nanoparticles efficiently modulated the immune suppressive TME and generated anti-cancer immune memory effect, resulting in killing primary tumors and distant tumors at the same time. The idea of cryo-immune modulation with cold responsive nanoparticles has been proposed with previous reports. The benefits of cryo treatment in immune modulation have proved in recent studies. Thus, various combinational immunotherapy approaches also have been reported. Specific advantages of using cold-responsive nanoparticles are not fully convincible.

Re: As discussed in the Introduction section, we do agree with the reviewer that cryosurgery has been reported for immune modulation or in combination with other therapies for enhancing the destruction of localized tumors. However, no study has been reported to show the abscopal effect to effectively destroy distant and metastatic tumors of cryosurgery either alone or in combination with other therapeutic strategies. In addition, no work has been published to combine cryo-immune modulation with cold responsive nanoparticles. With cold-responsive nanoparticle for co-delivery of chemotherapy drug and anti-PD-L1 immunosuppressive checkpoint to combine with cryosurgery, this work is the first to show that cryoimmunotherapy can be potentiated to destroy not only localized primary tumor, but also distant (Figs. 6-7) and metastatic tumors (Fig. 8). This is a major advance in the field of cryosurgery, and we hope the reviewer agrees that the advantage of using cold-responsive nanoparticles for cold-triggered drug and gene delivery to potentiate cryoimmunotherapy is evident and of great significance, as cancer metastasis is major cause of most cancer-related mortality. We have made this clearer in the first two paragraphs on page 4 (i.e., at the end of the Introduction section).

(Introduction)

It is not clear if the ultrasonography limitation is the main reason for the incomplete tumor destruction and cancer recurrence.

Re: Although more clinical studies are needed to find out if the ultrasonography limitation is the main reason for the incomplete tumor destruction and cancer recurrence after cryosurgery, it is clear that the ultrasonography limitation may lead to incomplete tumor destruction and cancer recurrence after cryosurgery, due to its inability to tell where the temperature is below -20 °C that is needed to ensure

cancer cell death. This was the concern that many of the previous work on combining cryosurgery with other therapeutic strategies were trying to address. We have made this clearer in the last paragraph on page 3.

(Design of nanocarriers)

Quite complex design with multiple components would be discouraging to translate the nanotechnology further.

Re: When designing the CRNPs, we used materials that are either FDA-approved for medical use (PLGA, PF127, DPPC) or naturally derived (chitosan) with good biocompatibility (Refs. 51-53), to facilitate the clinical translation of the CRNPs. This is now clarified in the last paragraph on page 4.

pNIPAM might not be a best choice for in vivo study with their reported potential toxicity.

Re: In our study, we do not observe any evident toxicity of our pNIPAM-BA-containing CRNPs when compared to the group with PBS injection, per the analysis of hemolysis (Fig. 4f), histology examination of the major organs (Supplementary Figs. 23-24), and assessment of blood ALT and AST (Supplementary Fig. 25). Therefore, our CRNPs with pNIPAM-BA is promising in terms of in vivo safety, although we do agree that more data are needed to definitely confirm the safety of the CRNPs.

It is not clear that the synergistic effect of Cryo + irinotecan + PD-L1 silencing siRNA combination therapeutics. It might be difficult to discriminate the synergistic combination effects in each addition of therapeutics.

Re: Per the reviewer's advice to assess the synergistic effect of Cryo + irinotecan + PD-L1 silencing siRNA combination therapeutics, we conducted more experiments to compare the tumor attacking capability of CPT&siR CRNPs (either with or without cold treatment) with that of CPT CRNPs alone, siR CRNPs alone, and the simple addition of the effect (i.e., additive effect) of the CPT CRNPs alone and siR CRNPs alone. The data are shown in the new Supplementary Fig. 10. In the absence of cold treatment, co-delivery of CPT and siR in the CPT&siR CRNPs has no synergistic effect and leads to 30.7% of cancer cell death, which is less than the sum value (39.0%, i.e., for the simple additive effect) of the antitumor effect of CPT CRNPs alone and siR CRNPs alone (Supplementary Fig. 10a-b). Importantly, when cold treatment is applied, the synergistic effect of CPT and siR co-delivered with CPT&siR CRNPs is evident (Supplementary Fig. 10c-d), showing significantly higher percentage of cancer cell death (80.5%) than the sum value (61.4%, i.e., for the simple additive effect) of CPT CRNPs alone and siR CRNPs alone (labeled in dotted box). The info is now added in 1st paragraph on page 9.

It is unclear that the use of Chitosan-PF127 and NaCl in the system.

Re: In this work, the use of chitosan-PF127 is to make the CRNPs with the capability of cancer cell targeting via chitosan that has been shown to target cancer cells and tumor in the literature (Ref. 52). This is further confirmed by the cellular uptake experiment of this work: CRNPs are internalized into 92.8% of EO771 cancer cells while only ~0.9% of normal lymphocytes can take up the CRNPs (Supplementary Fig. 4) and preferentially accumulate in tumor in vivo (Fig. 4a-b, d-e). This is now clarified at the end of the 1st paragraph of Results and Discussion on page 5 and when describing the data shown in Supplementary Fig. 4 in the 3rd paragraph on page 6 and in Fig. 4a-b, d-e in the 2nd paragraph on page 9 of this revised manuscript.

As for NaCl, it is encapsulated in the CRNPs for achieving endo/lysosomal escape. As shown in Fig. 2e and Supplementary Fig. 5, after cold treatment (CPT&Cy5-siR CRNPs+C), co-location of Cy5 and endo/lysosomes is decreased with an evident separation of red (Cy5-siR in CRNPs) and green (endo/lysosomes) fluorescence signals, indicating successful cold-triggered escape of Cy5-siR from endo/lysosomes into the cytosol. However, cold treatment on the group (CPT&Cy5-siR CRNPs+C, no NaCl) does not induce an evident separation between the red fluorescence of Cy5-siR and the green fluorescence of endo/lysosomes. The possible mechanism for the NaCl-mediated endo/lysosomal escape is now discussed in the last 4 lines of the 3rd paragraph on page 6 and the first 2 lines on page 7 of this revised manuscript.

(Therapy)

The role of irinotecan is unclear.

Re: Irinotecan (CPT) is a clinically used chemotherapy drug for breast cancer. In this study, we used it for enhancing immunogenic cell death (ICD) together with cryosurgery. As shown in Fig. 3a, combination of CPT&siR CRNPs with cryosurgery can induce a higher percentage of DAMPs when

compared to cryosurgery alone, CPT CRNPs alone, or siR CRNPs alone. In the in vitro T cell-mediated cancer cell attacking experiment, CPT&siR CRNPs with cold treatment causes higher percentage of cancer cell death than CPT CRNPs alone, siR CRNPs alone, or the additive effect of CPT CRNPs alone and siR CRNPs alone, either with/without cryosurgery (Supplementary Fig. 10). Furthermore, in the in vivo experiment, mice treated with cryosurgery and CPT&siR CRNPs (i.e., ICIE) show higher percentage of activated immune cells against primary, distant, and metastatic tumors than mice treated with CPT CRNPs alone or siR CRNPs alone either with/without cryosurgery. All these results show that CPT is necessary for inducing a strong immunogenic cell death (ICD) together with cryosurgery. The role of irinotecan is now clarified in the 1st paragraph of Results and Discussion on page 4 and when discussing Fig. 3a on page 7 and Supplementary Fig. 10 on page 9 of this revised manuscript.

Are those black regions necrotic areas or traces of probe insertion in Fig 7c?

Re: Sorry for the confusion! The black regions are a result of skin wound that often occurs in the nipple area due to orthotopic tumor growth. In our study, we used the orthotopic breast cancer model for obtaining the data shown in Fig. 7c. We have added this information to the Fig. 7c legend. We want to thank the reviewer again for all the insightful comments!

REVIEWERS' COMMENTS

Reviewer #1 (Remarks to the Author):

In this revised manuscript, the authors provided additional description and information to clarify the experiment procedures and data analysis. New experimental data support the experimental design and conclusions. Overall, the authors fully addressed my concerns and comments.

Reviewer #2 (Remarks to the Author):

In the response letter, corresponding author and coauthors have correctly answered my question, I am satisfied with their feedback to my comments. Therefore, the revised version can be accepted without further modification.

Reviewer #4 (Remarks to the Author):

Authors address most of comments from my review. One thing to consider would be additional discussion for potential options for temperature sensitive polymers that can replace pNIPAAm.

Point-by-point response to reviewers' comments

Only Reviewer #4 has one remaining comment, which together with our reply to the comment is given below:

Reviewer #4 (Remarks to the Author):

Authors address most of comments from my review. One thing to consider would be additional discussion for potential options for temperature sensitive polymers that can replace pNIPAAm.

Re: Per the reviewer's advice, we looked further into the literature on temperature sensitive polymers. Unfortunately, nearly all work in the biomedical field on other temperature sensitive polymers reported in the literature is focused on developing such polymer with a lower critical solution temperature (LCST) above room temperature, which is not applicable to this work on cold temperature (i.e., below room temperature) applications. Furthermore, our extensive data (Fig. 4f and Supplementary Figs. 22-25) show excellent biocompatibility of the cold responsive nanoparticles containing pNIPAAm-BA. Thus, we feel there is no need to add the discussion for potential options for temperature sensitive polymers that can replace the pNIPAAm-BA in this work.